**Analysis**

# The Cell Tracking Challenge: 10 years of objective benchmarking

Martin Maška [1], Vladimír Ulman [1,2], Pablo Delgado-Rodriguez [3,4], Estibaliz Gómez-de-Mariscal [3,4,5], Tereza Nečasová [1], Fidel A. Guerrero Peña [6,7], Tsang Ing Ren [6], Elliot M. Meyerowitz [8], Tim Scherr [9], Katharina Löffler [9], Ralf Mikut [9], Tianqi Guo [10], Yin Wang [10], Jan P. Allebach [10], Rina Bao [11,12], Noor M. Al-Shakarji [12], Gani Rahmon [12], Imad Eddine Toubal [12], Kannappan Palaniappan [12], Filip Lux [1], Petr Matula [1], Ko Sugawara [13,14], Klas E. G. Magnusson [15], Layton Aho [16], Andrew R. Cohen [16], Assaf Arbelle [17], Tal Ben-Haim [17], Tammy Riklin Raviv [17], Fabian Isensee [18,19], Paul F. Jäger [19,20], Klaus H. Maier-Hein [18,21], Yanming Zhu [22,23], Cristina Ederra [24], Ainhoa Urbiola [24], Erik Meijering [22], Alexandre Cunha [7], Arrate Muñoz-Barrutia [3,4], Michal Kozubek [1,25] ✉ & Carlos Ortiz-de-Solórzano [24,25] ✉

The Cell Tracking Challenge is an ongoing benchmarking initiative that has become a reference in cell segmentation and tracking algorithm development. Here, we present a significant number of improvements introduced in the challenge since our 2017 report. These include the creation of a new segmentation-only benchmark, the enrichment of the dataset repository with new datasets that increase its diversity and complexity, and the creation of a silver standard reference corpus based on the most competitive results, which will be of particular interest for data-hungry deep learning-based strategies. Furthermore, we present the up-to-date cell segmentation and tracking leaderboards, an in-depth analysis of the relationship between the performance of the state-of-the-art methods and the properties of the datasets and annotations, and two novel, insightful studies about the generalizability and the reusability of top-performing methods. These studies provide critical practical conclusions for both developers and users of traditional and machine learning-based cell segmentation and tracking algorithms.

The field of automated cell tracking has contributed extremely valuable tools to life scientists with which to conduct their research[1–3]. However, the emergence of technical developments that improve the resolution[4], dimensionality[5], extent and throughput[6] of optical microscopes demands new, improved tracking algorithms. Furthermore, the fast evolution of machine learning[7] is changing the way cell tracking is performed, as deep neural networks rapidly replace classical image analysis methods. These models provide impressive results while posing their own share of challenges related to their training strategies, quality and quantity of available training data, parametrization, and generalization.

The Cell Tracking Challenge (CTC) (http://celltrackingchallenge. net) is an ongoing initiative that promotes the development and objective evaluation of automated cell tracking algorithms. Launched in 2013 under the auspices of the 10th IEEE (Institute of Electrical and Electronics Engineers) International Symposium on Biomedical Imaging (ISBI), the CTC provides developers with a rich and diverse annotated dataset

repository of multidimensional time-lapse microscopy videos along with objective measures and procedures to evaluate their algorithms. These highly valuable resources are freely available to the scientific community for use in their research.

In 2014 the first report was published[8], describing the CTC submission and evaluation procedures and presenting the analysis of the results submitted by six participants for a repository containing eight datasets. In 2017 an in-depth analysis of 21 algorithms was published[9], based on the segmentation and tracking results submitted for 13 datasets. From the results presented, we concluded that the methods that used contextual (that is, spatial and temporal) information and those few at the time that followed learning strategies outperformed the more conventional methods. Notably, the state-of-the-art U-Net[10] architecture was among the top-performing approaches for cell segmentation in several contrast-enhanced datasets. It was also notable that completely unsupervised tracking methods were still a distant dream. The optimal solutions remained dataset specific due to the complexity and diversity of the datasets. Moreover, most proposed methods were still inadequate for low signal-to-noise ratio videos or for tracking cells with complex shapes or textures. Large three-dimensional (3D) datasets, such as those of developing embryos, were identified as extremely challenging due to the high number and density of cells, as well as the computational requirements of their processing.

Since 2017, the CTC has received a significant number of new submissions and has addressed many of the challenges previously identified in the field of automated cell tracking, as described in the following sections.

## Results

### Datasets

The CTC dataset repository has been extended from 13 datasets in 2017 to 20 datasets. The new datasets consist of two-dimensional (2D) epi-fluorescence time-lapse videos of human hepatocarcinoma-derived cells expressing a fusion yellow fluorescent protein (YFP)-TIA-1 protein (Fig. 1a); 2D bright-field time-lapse videos of mouse hematopoietic (Fig. 1b) or muscle (Fig. 1c) stem cells in hydrogel microwells; 3D time-lapse videos of green fluorescent protein (GFP)-actin A549 lung cancer cells (Fig. 1d) and their computer-generated counterparts (Fig. 1e) displaying prominent, highly dynamic filopodial protrusions; and mesoscopic videos (imaged across several millimeters at video frame rate) of developing *Tribolium castaneum* embryos available as 3D cartographic projections (>10 GB per sequence) (Fig. 1f) or as complete 3D datasets (>100 GB per sequence) (Fig. 1g). Supplementary Table 1 provides a technical description of all of the datasets and Fig. 2 contains a summary of the main quality properties of the datasets.

### Reference annotations

The CTC provides two measure-specific reference annotations: segmentation annotations consisting of cell instance masks, which outline individual cell regions, and tracking annotations consisting of cell markers interlinked between frames to form lineage trees. The reference annotations can be classified into three types based on their source and how they were generated. For synthetic datasets, generated using in-house developed software[11,12], the segmentation and detection and/or tracking reference annotations are the exact, simulated digital cell phantoms prior to the addition of distorting noise and blur. For real datasets, we distinguish between a gold standard reference corpus (in short, gold truth) and a novel silver standard reference corpus (in short, silver truth). The gold truth is obtained by taking a majority opinion between three experts. The segmentation gold truth offers limited cell instance coverage (17.8% on average; Supplementary Data Tabs 1 and 2) due to the labor-intensive nature of manual annotations. The detection and tracking gold truth offers complete cell instance coverage, except in large embryonic datasets, where only

a selected part of the embryo is covered. The silver truth consists of computer-generated segmentation annotations obtained by fusing the results of high-performing benchmarked methods over the training sequences, using the detection gold truth to drive the fusion process (see 'Silver standard reference annotation' in Methods). This silver truth improves the cell instance coverage (99.1% on average), providing the participants with a larger set of annotated cell instances that can be used, for example, to train deep learning models. Supplementary Data Tabs 1 and 2 contain the coverage of both gold truth and silver truth annotations for all datasets. Note that 100% coverage was not attainable because even the best-performing benchmarked methods did not always detect and segment all cell instances in a particular video. All reference annotations are publicly available for the training datasets but are kept secret for the test datasets. This helps prevent overfitting by not enabling the methods to be tuned specifically for the test data. Thus, it ensures that the performance of the methods is evaluated based on their ability to generalize rather than their ability to memorize the training data.

### Participants, algorithms and handling of submissions

The CTC has witnessed a remarkable increase in participation since the time of our last report in 2017 (ref. 9). The number of participating teams has increased from 16 to 50, representing 19 countries. The number of benchmarked algorithms has also increased from 21 to 89. All submissions, consisting of labeled segmentation masks and structured text files with cell-lineage graphs in the case of tracking results, followed standardized naming conventions and were verified by the CTC organizers using the provided executable versions of the algorithms. A complete list of the participants' segmentation and tracking algorithms can be found in Supplementary Data Tabs 3 and 4, and a global overview of the strategies and techniques used is presented in Fig. 3. Approximately one-third of segmentation methods use a separate detection step first (DetSeg) instead of segmenting the objects directly (Seg). Regarding the tracking task, we confirm the overall dominance of methods in which linking is based on a prior per-frame segmentation (SegLnk or DetSegLnk) over those in which the linking part is based on per-frame detection only (DetLnkSeg) or simultaneous segmentation and linking (Seg&Lnk).

### Technical performance of the submitted algorithms

The CTC has two benchmarks: the Cell Tracking Benchmark (CTB) and the Cell Segmentation Benchmark (CSB). The CTB, which has been active since the inception of the CTC, evaluates the segmentation (SEG) and tracking (TRA) accuracy of the submitted methods. The CSB, introduced in 2019, focuses on segmentation (SEG) and detection (DET) accuracy, without considering the linking of cells over time. All these evaluation measures are described in Methods (in the 'Quantitative performance criteria' section).

The scores of all CSB and CTB submissions received before 1 June 2022 can be found in Supplementary Data Tabs 5–9 (CSB) and Tabs 10–14 (CTB). Figure 4a shows the SEG and DET performance of the top-3 CSB methods, along with the overall performance measure (OP$_{CSB}$), calculated as the arithmetic mean of both measures. Likewise, Fig. 4b shows the SEG and TRA performance of the top-3 CTB methods, along with the overall performance measure (OP$_{CTB}$). To globally rank the methods, we computed the weighted number of occurrences of each method or its generalizable version (labeled with an asterisk, see 'Generalizability study') in the top-3 positions of the CSB and CTB leaderboards (Fig. 4). We assigned 1, 2 or 3 points for each top-3, top-2 and top-1 occurrence, respectively. Based on this calculation, the top-3 CSB methods are CALT-US (*) (ref. 13), KIT-GE (3) (ref. 14), and (sharing third place) DKFZ-GE[15], KIT-GE (4) (ref. 16) and KTH-SE (1) (ref. 17), and the top-3 CTB methods are KIT-GE (3), KIT-GE (4) and KTH-SE (1). A description of these methods is given in the 'Top-performing Algorithms' section in Methods (and on the challenge website).

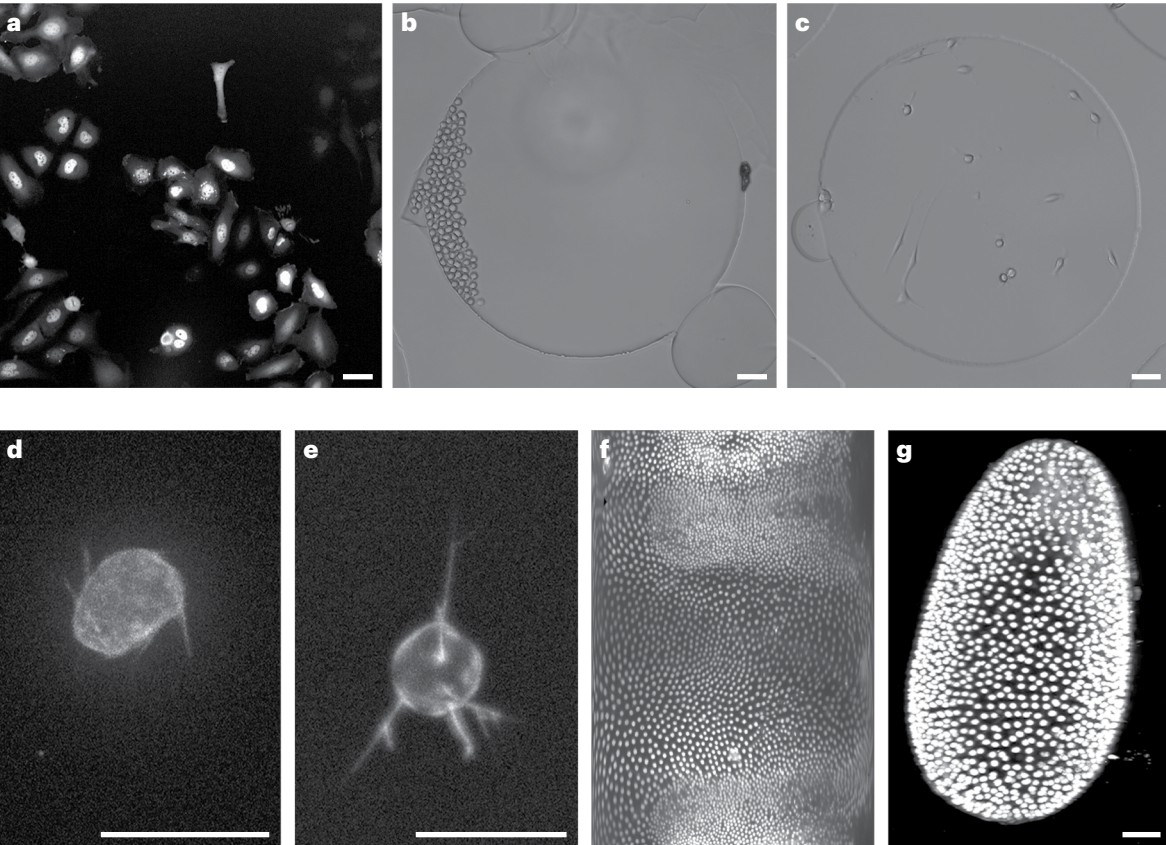

**Fig. 1 | CTC datasets added after 2017. a**, Fluo-C2DL-Huh7. **b**, BF-C2DL-HSC. **c**, BF-C2DL-MuSC. **d**, Fluo-C3DH-A549. **e**, Fluo-C3DH-A549-SIM. **f**, Fluo-N3DL-TRIC (due to the cartographic post-production of this dataset, the spatial resolution varies with position in the image between 0.10 and 0.76 µm per pixel; thus, a fixed-size bar is inappropriate for this dataset). **g**, Fluo-N3DL-TRIF. For the definitions of the dataset names please see the Fig. 2 legend. Scale bars: **a–c,g**, 50 µm; **d,e**, 20 µm.

Globally, the evolution of the CSB scores obtained by the best-performing methods from 2017 to June 2022 is given in Extended Data Fig. 1a, which shows clear improvement in both detection and segmentation on most datasets, with particularly impressive improvements on two of the most complex datasets (Fluo-C2DL-MSC and Fluo-N3DL-DRO). In summary, even if the cell detection task seems nearly solved for most datasets, the segmentation task still requires further attention for some of the old (Fluo-C2DL-MSC, Fluo-C3DL-MDA231, Fluo-N3DL-DRO and PhC-C2DL-PSC) and new datasets. Looking at the evolution of the CTB segmentation scores (Extended Data Fig. 1c), there is also significant improvement in most datasets but more work needs to be done to improve the segmentation and tracking performance in the same datasets that have been mentioned for the CSB.

**Image quality versus algorithm performance**
We analyzed the relationship between the technical performance values obtained by the participants (Supplementary Data Tabs 5–9 and Tabs 10–14) and the quality of the datasets listed in Fig. 2. As described in the 'Statistical analysis' section of Methods, we calculated the Spearman's rank correlation between each numerical quality measure of the dataset and the performance of all competing algorithms. The analysis was conducted globally, that is, considering all datasets, per data modality, and individually. The results are presented in Supplementary Figs. 1–40.

Globally, we discovered that only the cell overlap (Ove) showed correlation (moderate, rho = 0.4) with the segmentation performance (SEG) of the algorithms (Supplementary Fig. 34). This result, besides pointing at other cofactors that may work along with Ove, indicates that the cells that do not dramatically change their shape, or show a moderate motility, are easier to segment than those with high shape variability or high motility.

Looking at the correlations per modality, strong correlations were found between the performance of the methods on Fluo-2D datasets and the signal-to-noise ratio (SNR; positive for TRA, Supplementary Fig. 4), resolution (Res; negative for TRA, Supplementary Fig. 20), shape (Sha; positive for SEG, Supplementary Fig. 22) and mitotic division rate (Mit; positive for TRA, Supplementary Fig. 40). The positive effect of high SNR and regular cell shape (high Sha) could be expected. The counterintuitive benefit of a low Res for tracking can be explained by the negative effect of the low performance values obtained for two complex datasets, Fluo-C2DL-MSC and Fluo-C2DL-Huh7, which have relatively high Res levels but which are plagued by irregular cell shape (low Sha), high photobleaching (high change in cell signal intensity over time, that is, high Cha), low SNR and low contrast ratio (CR) (Fluo-C2DL-MSC), high levels of signal heterogeneity (both inside and between cells, that is, $Het_i$ and $Het_b$, respectively) and low Mit (both Fluo-C2DL-MSC and Fluo-C2DL-Huh7). These negative factors clearly outweigh the benefits of their relatively high Res. Regarding the Fluo-3D datasets, the high differences between datasets is reflected in the moderate correlations between the performance of the methods and Res (positive for SEG and TRA, Supplementary Figs. 18 and 20), Sha (negative for SEG, Supplementary Fig. 22), and the spacing between cells (Spa; positive for SEG Supplementary Fig. 26).

Bright-field performance values correlate with SNR (positive for SEG and TRA, Supplementary Figs. 2 and 4), CR (negative for SEG and TRA, Supplementary Figs. 6 and 8), $Het_i$ (negative for SEG and TRA, Supplementary Figs. 10 and 12), $Het_b$ (negative for SEG and TRA, Supplementary Figs. 14 and 16), Res (negative for SEG, Supplementary

| Name | SNR | CR | Het$_i$ | Het$_b$ | Res | Sha | Spa | Cha | Ove | Mit | Syn | Ent/Leav | Apo | Deb |
|---|---|---|---|---|---|---|---|---|---|---|---|---|---|---|
| BF-C2DL-HSC | 2.01 | 0.86 | 3.76 | 0.47 | 271 | 0.82 | 5.73 | 0.01 | 0.68 | 0.16 | N | N | N | N |
| BF-C2DL-MuSC | 1.09 | 0.95 | 17.26 | 0.97 | 944 | 0.57 | 77.74 | 0.00 | 0.57 | 0.02 | N | N | N | N |
| DIC-C2DH-HeLa | 0.45 | 1.00 | 32.59 | 1.42 | 12 093 | 0.70 | 7.05 | 0.43 | 0.91 | 0.02 | N | Y | Y | Y |
| Fluo-C2DL-Huh7 | 9.54 | 8.63 | 1.10 | 0.71 | 6 006 | 0.65 | 13.57 | 0.07 | 0.92 | 0.23 | N | Y | N | Y |
| Fluo-C2DL-MSC | 3.18 | 1.56 | 1.04 | 0.77 | 14 349 | 0.27 | 57.35 | 84.42 | 0.73 | 0.01 | N | Y | N | N |
| Fluo-C3DH-A549 | 16.15 | 3.23 | 0.37 | NA | 75 745 | 0.37 | NA | 0.50 | 0.89 | NA | N | N | N | N |
| Fluo-C3DH-H157 | 20.69 | 2.63 | 0.54 | 0.48 | 497 671 | 0.43 | 186.66 | 4.45 | 0.83 | 0.00 | N | Y | N | N |
| Fluo-C3DL-MDA231 | 11.32 | 5.02 | 1.07 | 0.26 | 1 306 | 0.48 | 14.48 | 4.44 | 0.68 | 0.17 | N | Y | N | N |
| Fluo-N2DH-GOWT1 | 4.92 | 9.99 | 0.89 | 0.85 | 3 268 | 0.84 | 56.29 | 0.01 | 0.92 | 0.07 | N | Y | N | Y |
| Fluo-N2DL-HeLa | 22.68 | 1.02 | 0.35 | 0.61 | 610 | 0.83 | 13.15 | 2.63 | 0.88 | 1.45 | N | Y | Y | Y |
| Fluo-N3DH-CE | 8.06 | 3.72 | 0.64 | 0.27 | 7 866 | 0.46 | 2.61 | 0.21 | 0.72 | 1.74 | Y | N | N | N |
| Fluo-N3DH-CHO | 7.27 | 6.90 | 0.97 | 0.35 | 19 921 | 0.29 | 33.39 | 0.02 | 0.88 | 0.06 | N | Y | Y | N |
| Fluo-N3DL-DRO | 2.25 | 3.12 | 0.31 | 0.19 | 1 832 | 0.62 | 7.18 | 1.18 | 0.67 | 1.05 | N | N | N | N |
| Fluo-N3DL-TRIC | 7.18 | 2.16 | 0.29 | 0.33 | 299 | 0.83 | 6.68 | 1.62 | 0.85 | 0.43 | Y | N | N | N |
| Fluo-N3DL-TRIF | 12.48 | 7.11 | 0.26 | 0.18 | 7 558 | 0.85 | 5.93 | 14.38 | 0.77 | 0.89 | N | N | N | Y |
| PhC-C2DH-U373 | 2.90 | 1.10 | 13.03 | 0.87 | 4 387 | 0.55 | 104.45 | 0.02 | 0.91 | 0.00 | N | Y | N | Y |
| PhC-C2DL-PSC | 3.12 | 1.44 | 0.87 | 0.40 | 143 | 0.59 | 4.87 | 0.08 | 0.91 | 1.99 | N | Y | N | Y |
| Fluo-C3DH-A549-SIM | 6.06 | 1.64 | 0.52 | NA | 59 587 | 0.33 | NA | 1.05 | 0.94 | NA | N | N | N | N |
| Fluo-N2DH-SIM+ | 6.35 | 1.23 | 0.96 | 0.48 | 1 809 | 0.73 | 19.24 | 0.18 | 0.89 | 0.49 | N | Y | N | N |
| Fluo-N3DH-SIM+ | 5.30 | 1.24 | 1.07 | 0.41 | 38 388 | 0.76 | 16.35 | 0.17 | 0.86 | 0.49 | N | Y | N | N |

Easy ⟶ Difficult

**Fig. 2 | Quantitative and qualitative properties of the test datasets.** For individual datasets, the columns show their numerical quality measures: signal-to-noise ratio (SNR), contrast ratio (CR), heterogeneity of the signal intensity inside the cells (Het$_i$) and between the cells (Het$_b$), resolution (Res), shape (Sha), spacing between cells (Spa), change in cell signal intensity over time (Cha), overlap (Ove) and mitotic division rate (Mit). The remaining columns list qualitative observations of various features, such as the presence of synchronous cell divisions (Syn), cells entering or leaving the field of view (Ent/Leav), apoptotic cells (Apo) or debris (Deb). For each quantitative property, the computed values are first filtered for outliers, that is, values more than 1.5-fold the interquartile range below the first quartile, or above the third quartile of the data. The remaining values are linearly mapped onto a green-yellow-red color scale to indicate the a priori level of complexity. (The outliers are shown with the darkest green and the darkest red backgrounds, located before and after the white vertical bars on the color key.) These values were computed using the methodology established in 2017 (see the 'Dataset properties' section in Methods). Dataset names: 2D, two dimensional; 3D, three dimensional; A549, human lung adenocarcinoma cells; BF, bright-field; C, cytoplasmic staining; CE, *Caenorhabditis elegans;* CHO, Chinese hamster ovarian cells; DIC, differential interference contrast; DRO, *Drosophila melanogaster;* Fluo, fluorescence; GOWT1, mouse embryonic stem cells; H, high resolution; H157, human oral squamous cell carcinoma cells; HeLa, Henrietta Lacks human uterine cervical carcinoma immortalized cells; HSC, mouse hematopoietic stem cells; Huh7, human hepatocarcinoma-derived cells; L, low resolution; MDA231, human breast metastatic adenocarcinoma lines; MSC, rat mesenchymal stem cells; MuSC, mouse muscle stem cells; N, nuclear staining; PhC, phase contrast; PSC, pancreatic stem cells; TRIC, *Tribolium castaneum* (cartographic projection); TRIF, *Tribolium castaneum* (full 3D volume); SIM, simulated cells; SIM+, second-generation simulated cells; U373, human glioblastoma–astrocytoma cells.

Fig. 18), Sha (positive for SEG, Supplementary Fig. 22), and Spa (negative for SEG, Supplementary Fig. 26). Most of these correlations could be expected, except for the counterintuitive effect of Res and Spa. This could be explained by the fact that in one of the two bright-field datasets (BF-C2DL-HSC), the negative effect of the low Res and Spa, compared with the other dataset, BF-C2DL-MuSC, seems to be lower than the benefits of a lower Het$_b$, uniform shape (high Sha) and high Ove. Regarding the phase contrast (PhC) datasets, the performance values correlate with CR (negative for SEG and TRA, Supplementary Figs. 6 and 8), Het$_i$ (positive for SEG and TRA, Supplementary Figs. 10 and 12), Het$_b$ (positive for SEG and TRA, Supplementary Figs. 14 and 16), Res (positive for SEG and TRA, Supplementary Figs. 18 and 20), Sha (negative for TRA, Supplementary Fig. 24), Spa (positive for SEG and TRA, Supplementary Figs. 26 and 28) and Cha (negative for SEG and TRA, Supplementary Figs. 30 and 32). These results are heavily influenced by the fact that the two types of phase contrast datasets available have strikingly different characteristics. This explains, for instance, the unexpected negative correlation found with CR, given that the dataset with higher CR values (PhC-C2DL-PSC) is more complex to analyze than PhC-C2DH-U373, due to the negative impact of other factors, most notably a significantly lower Res, Spa and higher Mit. Interestingly, the levels of heterogeneity (both Het$_i$ and Het$_b$), positively correlate with the performance of the methods for this modality, suggesting that the characteristic complex texture and halo-like artifacts of phase contrast images are beneficial for methods that are based on the recognition of patterns, as is the case for machine learning methods. Finally, no correlations were found for the only existing differential interference contrast (DIC) dataset, as could be expected due to the low number of elements ($n = 2$ videos) available for the analysis. Beyond these global and modality-specific results, other relevant observations (outside the scope and length of this paper) relating to the properties that affect the segmentation and tracking performances can be obtained from the distributions shown per dataset in Supplementary Figs. 1–40.

**Annotation quality versus algorithm performance**

We next analyzed the relationship between the performance of the algorithms and the quality of the available annotations. Figure 5 reports the quality of the gold truth segmentation (MSEG$_{GT}$), detection (MDET$_{GT}$) and tracking (MTRA$_{GT}$) annotations, and of the silver truth segmentation (SEG$_{ST}$) and detection (DET$_{ST}$) annotations. These quality parameters were calculated as explained in the Quality of Annotations and Human-level Performance section in the Methods. Note that these annotation quality measurements are not mutually comparable because the former assesses the difficulty of the manual annotation task itself (that is, how much the annotators agreed when manually annotating a particular video), whereas the latter assesses the quality of the fused computer-generated results.

We next looked at the correlation between the quality of the reference annotation parameters listed in Fig. 5, and the performance of the competing submitted algorithms (Supplementary Data Tabs 5–9 and Tabs 10–14). The complete set of results can be found in Supplementary Figs. 41–50. Globally, all three gold truth quality annotation parameters moderately correlate with the performance of the algorithms (Supplementary Figs. 42, 44 and 46), conveying the arguable expectation

**a**

**Strategies**

| Segmentation-only task [27] |
| --- |
| Segmentation (Seg) [16] |
| Detection → Segmentation (DetSeg) [11] |

| Segmentation and tracking task [62] |
| --- |
| Segmentation → Linking (SegLnk) [38] |
| Segmentation ↔ Linking (Seg&Lnk) [5] |
| Detection → Segmentation → Linking (DetSegLnk) [13] |
| Detection → Linking → Segmentation (DetLnkSeg) [6] |

**b**

**Techniques**

| | Detection |
| --- | --- |
| D1 | Thresholding |
| D2 | Peak localization |
| D3 | Machine learning |

| | Segmentation |
| --- | --- |
| S1 | Thresholding |
| S2 | Region growing |
| S3 | Machine learning |
| S4 | Energy minimization |

| | Linking |
| --- | --- |
| L1 | Label propagation |
| L2 | Nearest neighbor |
| L3 | Graph-based optimization |
| L4 | Contour evolution |
| L5 | Machine learning |

| | D1 | D2 | D3 | S1 | S2 | S3 | S4 | L1 | L2 | L3 | L4 | L5 |
| --- | --- | --- | --- | --- | --- | --- | --- | --- | --- | --- | --- | --- |
| Intensity | | | | 15 | 7 | | 5 | | | | 3 | |
| Boundary | | | | | | 3 | | | | | 2 | |
| Spatial statistics | | | | 4 | | | | | | | | |
| Spatiotemporal statistics | 3 | 2 | | | | | | | | | | |
| Distance | | | | | 5 | | | 1 | 14 | | | |
| Overlap | | | | | | | | 15 | 2 | | | |
| Motion analysis | | | | | | | | 6 | 1 | 1 | 1 | |
| Shortest path | | | | | | | | | | 7 | | |
| Minimum cost flow | | | | | | | | | | 5 | | |
| Probability | | | | | | | 1 | | | 6 | | |
| Multiple hypothesis | | | | | | | | | | 1 | | |
| Decision tree | | | 1 | | | 2 | | | | | | |
| U-Net variant | | | 23 | | | 38 | | | | | | |
| R-CNN variant | | | 4 | | | 4 | | | | | | |
| HRNet variant | | | 2 | | | 2 | | | | | | |
| Siamese tracker | | | | | | | | | | | 3 | |
| Graph neural network | | | | | | | | | | | | 1 |

**Fig. 3 | Taxonomy of the strategies and techniques used by the challenge participants. a**, A taxonomy of the cell segmentation and tracking strategies followed. **b**, Stratification of the detection, segmentation and linking techniques used by the benchmarked methods. The numbers in brackets are the number of submissions received for individual tasks and the number of submissions that followed a particular strategy. The numbers in the table indicate the number of submissions that use each technique.

that what is difficult for humans to do is also difficult for automated algorithms to solve. In the context of segmentation, there exists room for more consistent annotation, as indicated by the $MSEG_{GT}$ quality scores (Fig. 5). Therefore, increasing the consistency of the annotations should improve algorithm performance. Our per-modality look at the correlations confirm the global trend with different levels of strength, except for DIC, which could be partly due to the low number of datasets of this modality. Regarding the quality of the silver truth annotations, a strong or moderate global correlation of the quality parameters with $SEG_{ST}$ and $DET_{ST}$ was found (Supplementary Figs. 48 and 50). This is also expected given that the silver truth annotations were obtained as a combination of the best-performing methods, resulting in almost fully annotated datasets. Finally, modality-based and individual deviations from this rule were also found, in most cases due to a low number of datasets of the modality.

**Evolution of the segmentation and tracking paradigms**

We analyzed how different segmentation and tracking strategies (Fig. 3a), as well as individual detection, segmentation and linking techniques (Fig. 3b), relate to the technical performance of the bench-marked algorithms. Our analysis shows that the DetSeg strategies significantly outperform the Seg strategies for the datasets with heavily clustered cells such as DIC-C2DH-HeLa (Extended Data Fig. 2b). Indeed, the machine learning-based detection of individual cells turns out to be a crucial factor that reduces the number of under-segmentation and over-segmentation errors penalized by DET, as first demonstrated by

MU-CZ (2) and MPI-GE (CBG) (1) in two dimensions and three dimensions, respectively, in 2019 (Extended Data Fig. 3 and Supplementary Data Tabs 3 and 4). Nowadays, this detection-driven strategy dictates also the state of the art when analyzing embryonic datasets, as shown by the fact that the top three places in terms of DET are mostly occupied by detection-driven strategies (IGFL-FR, JAN-US, MPI-GE (CBG) (2), MPI-GE (CBG) (3), OX-UK and RWTH-GE (3)) for the Fluo-N3DH-CE, Fluo-N3DL-DRO, Fluo-N3DL-TRIC and Fluo-N3DL-TRIF datasets (Fig. 4a). In terms of segmentation performance, machine learning-based techniques globally outperform traditional thresholding-based and region-growing-based techniques. This holds for label-free microscopy datasets (Extended Data Fig. 4a–c), for which the establishment of appropriate handcrafted features and rules is generally more difficult than learning them autonomously using neural networks, and also for both the Fluo-2D (Extended Data Fig. 4d) and Fluo-3D (Extended Data Fig. 4e) datasets. Over time, one can observe a substantial improvement in the segmentation performance thanks to the introduction of self-configured neural networks (Extended Data Fig. 5 and Supplementary Data Tabs 3 and 4), such as the nnU-Net ('no new U-net') used in DKFZ-GE or NAS (neural architecture search) used in UNSW-AU, as well as multi-branch predictions used in KIT-GE (3) and KIT-GE (4). Finally, we have not found any statistically significant difference in the tracking performance (TRA) of machine learning-based and non-machine learning-based linking techniques across all datasets (Extended Data Fig. 6). Overall, over the 10 year existence of the CTC, one can observe a greater performance improvement of the rapidly evolving machine

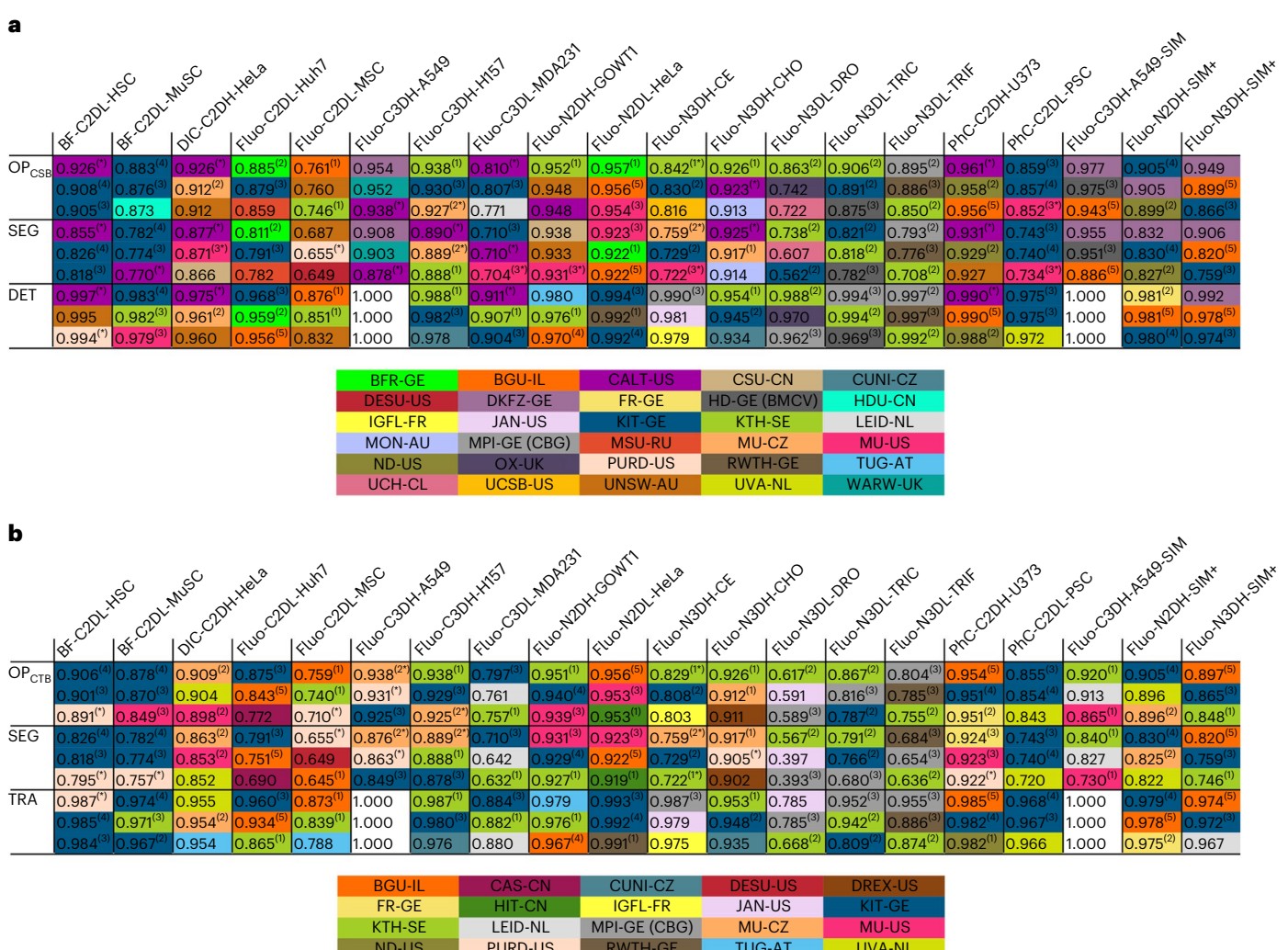

**Fig. 4 | Technical CTC leaderboards. a,b** Top-3 CSB (**a**) and CTB (**b**) leaderboards showing the top-performing methods for each of the available datasets. The numbers in parentheses indicate different algorithms submitted by the same group. Please see Supplementary Data Tabs 3 and 4 for the naming conventions and further details about the methods, and Supplementary Data Tabs 5–9 and Tabs 10–14 for a complete list of scores and a full ranking of the methods. For the definitions of the dataset names please see the Fig. 2 legend.

learning-based methods, compared to their machine learning-free competitors starting since 2019 in the case of DET (Extended Data Fig. 3) and SEG (Extended Data Fig. 5), and since 2021 in the case of TRA (Extended Data Fig. 7). This can be attributed not only to the introduction of the CSB itself, but also to the organization of three ISBI challenge editions with a fixed deadline.

### Biological performance of the submitted algorithms

The biologically inspired measures (see the 'Quantitative performance criteria' section in Methods) emphasize some aspects of the algorithm results that are of particular interest to the biologist, such as the ability of the algorithms to retrieve complete tracks (CT) or their large fractions (TF, track fractions), the accuracy of identifying division events to a tolerance of i frames (BC(i), branching correctness) and the accuracy of identifying complete cell cycles (CCA, cell cycle accuracy). A leaderboard based on these biological measures, as of 1 June 2022, is given in Fig. 6. The complete biological performance of benchmarked algorithms is available in Supplementary Data Tabs 15–20. The evolution of these scores for the best-performing methods from 2017 to June 2022 is shown in Extended Data Fig. 8.

Interestingly, the methods that provide the best biological performance, KIT-GE (3), KTH-SE (1) and KIT-GE (4), are also the ones that perform best in the technical aspects of the complete tracking tasks, even if not directly optimized for the biological measures. As in the case of the technical measures, there has been substantial progress since our 2017 report. This is particularly evident in the complex DIC-C2DH-HeLa dataset, the two mesoscopic datasets Fluo-N3DH-CE and Fluo-N3DL-DRO, and the simulated datasets Fluo-N2DH-SIM+ and Fluo-N3DH-SIM+. However, given that the methods are not directly optimized for these measures, much work is needed to improve the biological performance of the cell tracking algorithms, particularly in terms of CT and TF, which are essential for accurate and complete lineage tracking of embryonic datasets.

### Correlation between technical and biological measures

We have analyzed the relationship between the tracking (TRA) measure and the biologically inspired tracking measures (CT, TF, BC(i) and CCA), by means of the Spearman's rank correlation (see the 'Statistical analysis' section in Methods). The results can be found in Supplementary Figs. 51–56. Globally, a strong correlation between TRA and TF (rho = 0.698) and a moderate correlation between TRA and CT (rho = 0.608) were observed (Supplementary Fig. 51). We also learned that the relationship between TRA and TF is nearly linear, whereas it is more complex and non-linear between TRA and CT, given that retrieving

| Name | MSEG$_{GT}$ | | MDET$_{GT}$ | | MTRA$_{GT}$ | | SEG$_{ST}$ | DET$_{ST}$ |
|---|---|---|---|---|---|---|---|---|
| | Average | s.d. | Average | s.d. | Average | s.d. | | |
| BF-C2DL-HSC | 0.892 | 0.036 | 0.996 | 0.005 | 0.995 | 0.006 | 0.876 | 0.998 |
| BF-C2DL-MuSC | 0.843 | 0.026 | 0.994 | 0.003 | 0.993 | 0.003 | 0.821 | 1.000 |
| DIC-C2DH-HeLa | 0.784 | 0.066 | 0.965 | 0.044 | 0.965 | 0.044 | 0.896 | 0.997 |
| Fluo-C2DL-Huh7 | 0.876 | 0.027 | 0.947 | 0.036 | 0.938 | 0.035 | 0.849 | 1.000 |
| Fluo-C2DL-MSC | 0.769 | 0.047 | 0.971 | 0.013 | 0.969 | 0.014 | 0.795 | 0.948 |
| Fluo-C3DH-A549 | 0.859 | 0.020 | 1.000 | 0.000 | 1.000 | 0.000 | 0.918 | 1.000 |
| Fluo-C3DH-H157 | 0.924 | 0.009 | 0.996 | 0.004 | 0.991 | 0.005 | 0.921 | 1.000 |
| Fluo-C3DL-MDA231 | 0.742 | 0.048 | 0.939 | 0.009 | 0.935 | 0.010 | 0.762 | 0.997 |
| Fluo-N2DH-GOWT1 | 0.886 | 0.062 | 0.996 | 0.002 | 0.995 | 0.003 | 0.947 | 0.989 |
| Fluo-N2DL-HeLa | 0.904 | 0.035 | 0.987 | 0.002 | 0.987 | 0.002 | 0.936 | 1.000 |
| Fluo-N3DH-CE | 0.844 | 0.021 | NA | | NA | | 0.787 | 0.994 |
| Fluo-N3DH-CHO | 0.904 | 0.081 | 0.986 | 0.010 | 0.985 | 0.011 | 0.940 | 0.993 |
| Fluo-N3DL-DRO | 0.840 | 0.035 | NA | | NA | | 0.752 | 0.998 |
| Fluo-N3DL-TRIC | 0.902 | 0.004 | NA | | NA | | 0.856 | 1.000 |
| Fluo-N3DL-TRIF | 0.838 | 0.063 | NA | | NA | | 0.834 | 0.999 |
| PhC-C2DH-U373 | 0.836 | 0.143 | 0.994 | 0.005 | 0.992 | 0.006 | 0.933 | 0.999 |
| PhC-C2DL-PSC | 0.788 | 0.044 | 0.983 | 0.010 | 0.980 | 0.012 | 0.782 | 0.999 |
| Fluo-C3DH-A549-SIM | NA | | NA | | NA | | NA | NA |
| Fluo-N2DH-SIM+ | NA | | NA | | NA | | NA | NA |
| Fluo-N3DH-SIM+ | NA | | NA | | NA | | NA | NA |

Easy ▰▰▰▰▰▰▰▰▰▰▰▰▰▰ Difficult

**Fig. 5 | Quantitative properties of the reference annotations.** Quality of the human gold standard segmentation (MSEG$_{GT}$), detection (MDET$_{GT}$) and tracking (MTRA$_{GT}$) annotations, and of the fused silver standard segmentation (SEG$_{ST}$ and DET$_{ST}$) annotations, relative to the gold truth for all available datasets. Following the coloring scheme of Fig. 2, the color code in this table illustrates the relative difficulty of the annotation task, reflected by the level of agreement between annotators (gold standard annotations) and the quality of the silver standard annotations. See the 'Quality of annotations and human-level performance' section in Methods for an expanded explanation of how the values were calculated. For the definitions of the dataset names please see the Fig. 2 legend.

a high number of complete tracks requires a high technical tracking accuracy (TRA), but the opposite is not necessarily true. The same tendency can be observed for microscopy modalities (Supplementary Figs. 52–56) with strong (BF, DIC, Fluo-3D) or very strong (Fluo-2D and PhC) correlation between TRA and TF; and moderate (DIC), strong (BF, Fluo-2D and Fluo-3D), or very strong (PhC) correlations between TRA and CT. The correlations between TRA and BC(i) and TRA and CCA are moderate for PhC and strong for Fluo-3D, and show the same type of non-linear relationship as TRA and CT.

### Generalizability study

We evaluated the ability of the methods to provide competitive results across all datasets. To this end, we analyzed only methods that achieved a highly competitive OP$_{CSB}$ score, that is, an OP$_{CSB}$ score comparable to the third-ranked CSB method for at least one of the 13 included datasets, as of 15 January 2021. In this case, 'comparable score' meant that the difference between the OP$_{CSB}$ score of the method and that of the third-ranked CSB method was less than 1 standard deviation of the OP$_{CSB}$ scores of the three human annotations for a particular dataset. Furthermore, we reasoned that if the methods were trained on 13 training datasets, they would work well not only across the test datasets of these 13 datasets but also on unseen data types. With data types we refer to datasets different from the ones used for training.

The aim of the generalizability study was to compare the performance of the methods using six training configurations with different composition and extent: three individual per-dataset training approaches (producing an individual model for each dataset) exploiting gold truth; silver truth; and both gold and silver truth (GT + ST); versus three merged-all-datasets training approaches (producing a common model for all datasets) exploiting gold truth (allGT); silver truth (allST); and both gold and silver truth (allGT + allST). These six configurations multiplied by 13 datasets yielded a set of 78 results for each analyzed method.

In total nine teams participated in the generalizability study, each supplying all requested 78 results. Five of these groups also provided tracking results along with their segmentation results. To further evaluate the generalizability of these methods, we blindly applied the submitted solutions to unseen data types, namely to three datasets that were not used to train the methods (Fluo-C2DL-Huh7, Fluo-N2DH-SIM+ and Fluo-N3DH-SIM+). Extended Data Fig. 9 shows the performance of the top-3 most generalizable CSB (Extended Data Fig. 9a) and CTB (Extended Data Fig. 9b) methods, along with a more detailed look at the per-training configuration and per-dataset results of the top-1 CSB (CALT-US) (Extended Data Fig. 9c) and CTB (PURD-US[18]) (Extended Data Fig. 9d) methods. Supplementary Data Tabs 21–33 (CSB) and Tabs 34–42 (CTB) contain the results for all of the participants. Interestingly, the only non-machine learning algorithm that entered the study, KTH-SE (1), competed at the level of the best machine learning methods in the CTB category. This method was optimized with fluorescence data only, which shows a high degree of generalizability.

Extracting global conclusions from these results is complex, given that the behavior of the methods varies between datasets and to a lesser extent between algorithms. A general trend found is that, with different intensities and some differences between algorithms, the use of an extended per-dataset training configuration (silver truth or GT + ST) improves the performance of the algorithm compared with using only gold truth. Most of this increase is achieved by applying the dataset-specific silver truth alone. This improvement is small or moderate for most datasets, meaning that modern machine learning methods can produce competitive results using only the limited amount of training data contained in gold truth. Larger performance boosts (between 10% and 15% or higher in the case of PURD-US, Supplementary Data Tabs 34–42) can be seen for BF-C2DL-MuSC, DIC-C2DH-HeLa, Fluo-C3DL-MDA231, Fluo-N3DH-CE and PhC-C2DL-PSC. A common feature of these datasets is that their gold truth is difficult to obtain (medium–low MSEG$_{GT}$ scores in Fig. 5) and, except for Fluo-C3DL-MDA231, their gold truth annotation is very sparse (from 0.3% to 9%, Supplementary Data Tabs 1 and 2), emphasizing the importance of both the annotation quality and the coverage factor in the training process.

Furthermore, the improvement provided when training using merged-all-datasets configurations is comparable to the one obtained using per-dataset training, highlighting that machine learning models

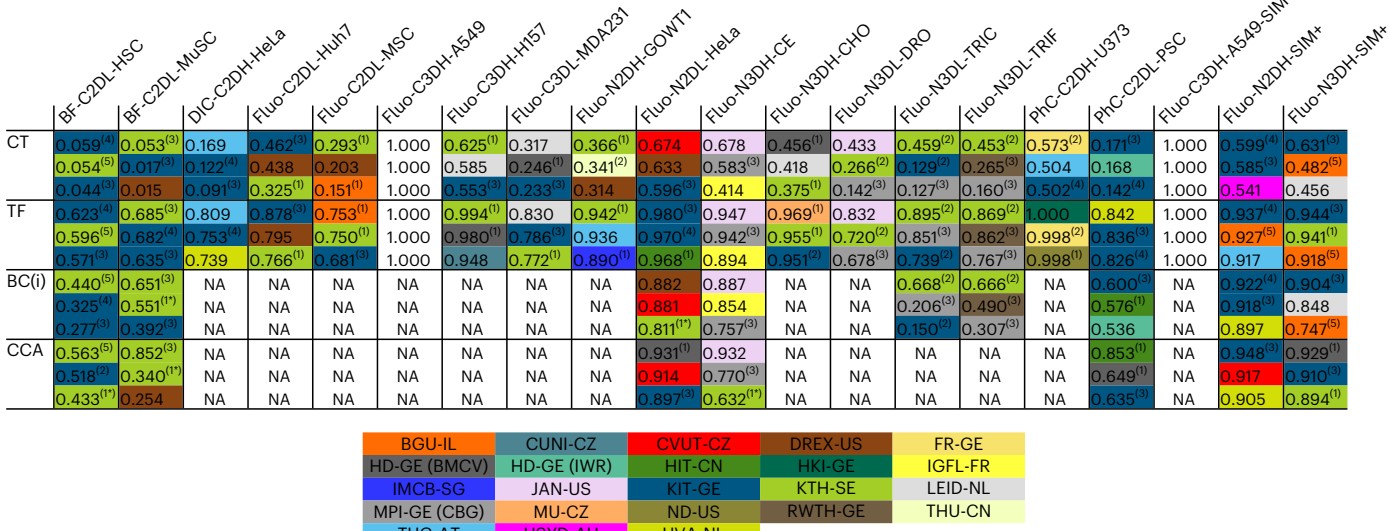

**Fig. 6 | Leaderboard of the top-3 performers in terms of biological measures.** Top-3 leaderboard showing the top-performing methods in terms of biological performance measures, for each of the available datasets. Please see Supplementary Data Tabs 3 and 4 for naming conventions and further details about the methods, and Supplementary Data Tabs 15–20 for a complete list of scores and a full ranking of the methods based on these measures. The numbers in parentheses indicate different algorithms submitted by the same group. For the definitions of the dataset names please see the Fig. 2 legend.

can indeed generalize from limited training data. An exception to this rule is Fluo-N3DH-CE, for which adding non-dataset-specific sequences for training produces worse results than using the silver truth alone. This may happen because the properties of this dataset differ greatly from the other datasets. Indeed, Fluo-N3DH-CE is the only real 3D dataset with approximately spherical objects included in the generalization study. Furthermore, this dataset has a significantly higher cell density (that is, low Spa in Fig. 2), higher Ove, higher Mit and a lower temporal sampling rate compared with the other fluorescent and non-fluorescent datasets. Because of these differences, a model trained with merged-all-datasets configurations is not as optimized for this dataset as a model specifically trained for it.

Regarding the performance of the most generalizable methods on data types not seen by their corresponding models during their training phase (Fluo-C2DL-Huh7, Fluo-N2DH-SIM+ and Fluo-N3DH-SIM+), we observed that the scores achieved are low, between 0.4 (Fluo-C2DL-Huh7) and 0.6 (Fluo-N2DH-SIM+ and Fluo-N3DH-SIM+). Initially, this suggested that current machine learning methods have little generalizability capacity. However, focusing on Fluo-C2DL-Huh7, we can see a similar performance to its more similar dataset (Fluo-C2DL-MSC). In fact, these are the only 2D fluorescent datasets with cytoplasmic staining, thus they are quite different from the rest of the datasets used in the study. This similarity explains why the methods achieve a similar degree of generalization on these two datasets. Likewise, the poor performance obtained for the synthetic datasets (Fluo-N2DH-SIM+ and Fluo-N3DH-SIM+) when using extended training configurations could be explained by the synthetic nature of these datasets, given that they have different image properties to real datasets, as already discussed for Fluo-N3DH-SIM+ and the most modality-similar datasets, Fluo-N3DH-CE and Fluo-N3DH-CHO, and can be affirmed also for Fluo-N2DH-SIM+ and Fluo-N2DH-GOWT1.

### Reusability efforts

Most deep learning-based methods do not perform well on datasets that are of a different type to those that they were trained on. Therefore, their reusability relies on the possibility to retrain these networks using new images. Indeed, fine-tuning of pre-trained models, such as those submitted to the challenge, can help to reduce the amount of new data that would be needed to train a model from scratch.

Thus, we proposed the following series of optional guidelines for the algorithm developers to make their workflows more reusable. First, the source code should be made available through a public repository (such as GitHub). Second, the repository should include clear instructions on how to initialize the model, load pre-trained weights, train it on new data and use it with new data. Third, all of the prerequisites to run the workflow must be clearly specified, and the authors should provide a way to easily install and load the model as part of their workflow. And last, for Python codes (commonly used for deep learning methods), creation of a Jupyter Notebook compatible with Google Colaboratory (Colab) is recommended, where new users can easily go through the steps mentioned in the second guideline.

Eleven participants followed the guidelines mentioned above. In particular BGU-IL (5) (refs. 19,20), CALT-US (*), DKFZ-GE, IGFL-FR (*) (ref. 21), KIT-GE (3), KIT-GE (4), MU-CZ (2*) (ref. 22), MU-US (3*) (ref. 23), MU-US (4*) (ref. 24) and PURD-US (*) followed all of the reusability guidelines. KTH-SE (1*), implemented in MATLAB, was also considered reusable. The Colab notebook implementations are available in the CTC GitHub repository (https://github.com/CellTrackingChallenge/2021-edition-available-colabs).

### Discussion

Since our last report[9] some important advances have occurred in the field, leading to significant improvements in the CTC, as discussed here.

The CTC's first and possibly most important news, which reflects the evolution in the field, is the overwhelming presence of machine learning, most notably, deep learning models for cell segmentation. This is reflected by the number of submitted algorithms that rely on machine learning for the segmentation (60 out of 89). More importantly, these models outperform traditional, non-machine learning-based methods, as four out of the five algorithms at the top of the CSB and CTB leaderboards (KIT-GE (3), KIT-GE (4), CALT-US (*) and DKFZ-GE) use deep learning-based segmentation strategies. However, a few algorithms based on traditional segmentation approaches remain at the top-1 of the leaderboards for one or more datasets, and have a performance at the level of the methods based on machine learning. This is the case for KTH-SE (1) and BGU-IL (1), which base their dominance on competitive segmentation and outstanding detection and linking approaches, and KTH-SE (2), which indeed

shows impressive segmentation and tracking performance for the embryonic datasets.

This fast replacement of traditional methods by machine learning-based methods is also observed globally, given that the performance of machine learning-based submissions has grown faster than that of traditional methods, with performance boosts coinciding with specific machine learning developments such as self-configured or multi-branched neural networks. Notably, most high-performing deep learning-based cell segmentation algorithms use variants of the popular U-Net architecture for detection and/or segmentation, with only a few using R-CNN[25] (region-based convolutional neural network) or HRNet (high-resolution net) variants[26]. For instance, DKFZ-GE uses a U-Net that automatically finds the optimal parameters for each dataset (nnU-Net); and BGU-IL (5) (ref. [19]) uses a recurrent neural network based on the combination of convolutional long short-term memory (ConvLSTM) layers and the U-Net. These examples show how models that evolved from the original U-Net architecture continue to be at the top in terms of competitiveness. This dominance of the U-Net for cell segmentation could be explained by its relative topological and conceptual simplicity. U-Net implements a bottom-up approach in which each pixel in the original image is semantically classified in a fine-grained fashion. Cell instances are then typically obtained by a clustering post-processing step. Thus, by combining shallow local details from the encoder and deep semantics from the decoder by the skip-connections and feature concatenation, this pixel-wise semantic segmentation facilitates fine localization of the cell boundaries.

A few methods are also pioneering the use of machine learning for cell linking. For example, this is the case for BGU-IL (5) (refs. [19,20]), which models the time-lapse sequence as a direct graph and uses a graph neural network (GNN) to extract the entire set of cell trajectories from the maximal paths in the graph. However, to date there are no remarkable differences in the performance of machine learning-based and traditional linking techniques. A limiting factor in the widespread use and improved performance of deep learning for tracking is the limited availability of densely annotated, fully tracked datasets. Because of this, traditional non-machine-learning based methods are still used by most groups. An interesting approach in this context is the recent development of algorithms that combine deep learning and traditional (for example, optimization[27]) principles.

Even fewer methods use deep learning to integrate cell segmentation and tracking. KIT-GE (4) is a noteworthy exception as it incorporates the linking step as one branch of a two-branched network model. The reason for this could be that, in the context of well-segmented datasets, no special linking approach seems necessary. This is in agreement with the literature, as most published workflows use a two-stage approach in which the segmentation part is optimized using data-driven objective metrics and the tracking part is tuned indirectly or even manually as a post-processing step. This practice limits the improvement of cell tracking, as the state-of-the-art tracking methods do not incorporate deep learning-based developments already being used in the fields of object tracking and motion estimation[28,29–32]. These methods have been used, for instance, for video super-resolution[33] and video rate conversion[34] but are yet to be explored in the context of cell tracking.

Understanding which parameters affect the performance of a cell tracking algorithm is a complex task, given that they are numerous and deeply interconnected. On the one hand, the quality of a time-lapse microscopy dataset heavily depends on the properties of the sample and the imaging and experimental setup. On the other hand, the performance of the algorithms also depends on the pre- and post-processing steps, the amount and quality of the training data, and the fabric of the algorithm itself. Consequently, it is not a surprise that our analysis identified only one factor, the cell overlap in consecutive frames (that is, a proxy of the variability of the cell morphology), that globally affects algorithm performance. However, a larger set of parameters

that are important for each modality or for each dataset have also been identified. This information should help biologists to optimize their imaging or experimental setup, and algorithm developers to address the issues that are more relevant for a specific dataset. Furthermore, our analysis confirms that the performance of the algorithms is directly related to the quality of the reference annotations. This is true for the gold standard and, in an even stronger way, for the silver standard reference annotations. This highlights the benefits of the increased coverage provided by the silver annotations and emphasizes the need for future efforts to improve the creation of the silver truth, given that improvements in the silver annotation result in direct improvements in the segmentation algorithms.

The CTC guidelines for standardizing input and output formats help to make algorithms easily reusable. In the case of deep learning approaches we have implemented guidelines to encourage participants to render pre-trained, easily reusable models. These guidelines promote open access to the source code, and clear instructions on how to retrain the models for new datasets and how to load pre-trained weights for transfer learning. To facilitate reusability, we recommend using Google Colab Notebooks, which provide a convenient open access tool for running code on an external server and provide free GPU access, for short training and test sessions. Although these reusability guidelines are currently optional, we plan to enforce them for all future challenge submissions. At the time of this report, 11 submitted algorithms, including all of the top-performing methods, comply with these reusability instructions, making it easier to transfer their methodologies to new laboratories or experiments.

As part of our efforts to improve the generalizability of the submitted algorithms, we conducted a study to evaluate how sensitive the methods are to changes in the training strategy. To facilitate generalization, we first produced silver standard reference segmentation annotations from the results submitted by the top performers. This is of particular interest for datasets with limited gold standard reference segmentation annotations. The results of our generalizability study are specific to each dataset and algorithm but we concluded that, in general, the most competitive methods do not require extensive data for training. That is, using a relatively small dataset-specific training set, these methods can provide similar results to those obtained by training with extra silver truth data from the same and other datasets. However, we also found that for certain datasets, particularly those with more difficult to obtain and sparse annotations (BF-C2DL-MuSC, Fluo-C3DL-MDA231, Fluo-N3DH-CE and PhC-C2DL-PSC DIC-C2DH-HeLa), these methods do benefit from the use of extended dataset-specific and, to a lesser extent, non-dataset-specific data. Furthermore, when we apply these pre-trained methods to unseen data types, their performance decreases significantly, indicating that they have difficulties generalizing to unseen data types involving different morphologies and/or imaging modalities. All together, these results highlight the practical importance of having pre-trained models that biologists can use to analyze new data types after fine-tuning on their own datasets.

Another important novelty of the challenge is the addition of new datasets that increase the diversity of the available training material. This includes light sheet microscopy videos of embryonic development, high-resolution real and synthetic confocal microscopy videos of highly protruding migrating cancer cells, and bright-field microscopy videos of proliferating stem cells. Dataset diversity is especially important in the context of deep learning models, which often struggle to analyze new datasets for which only a handful of cell images have been captured and which could benefit from transfer generation-based strategies[35]. These new datasets pose specific challenges. Light sheet microscopy is heavily used in developmental biology, wound repair and mechanobiology. Its growing use has led to a need for cell segmentation and tracking tools that can handle massive datasets. To facilitate this, we have extended our dataset repository to contain new *Tribolium castaneum* embryonic datasets in cartographic (20 GB size) and full 3D

tomographic versions (470 GB size). These extremely large datasets test the algorithms' accuracy and efficiency in handling large datasets. Interestingly, some submitted methods have achieved near annotator performance on these massive datasets (for example, KTH-SE (2) (refs. 17,36) and MPI-GE (CBG) (2) (ref. 37). The new real and synthetic videos of cancer cells with actin-stained protrusions are relevant for the study of mesenchymal cell migration and wound repair and require specific attention to accurately define cell boundaries in the presence of highly non-uniform and dynamic protruding extensions. The new mouse hematopoietic and muscle cells grown in hydrogel microwells provide efficient, high-throughput models for studying the effect of the microenvironment on stem cell fate, which is important for stem cell biology research. These datasets also present a specific technical challenge in detecting high mitotic activity.

An important aspect of this work is the establishment of the new segmentation-only CSB benchmark, in response to requests from researchers who wanted to benefit from our benchmarking efforts for the task of segmenting cells without addressing the tracking part of the problem. However, as shown by the global evolution of the scores since our 2017 report, cell segmentation and tracking continue to require new, more refined algorithms to improve both the technical and biological measures and, importantly, to improve the generalizability of the deep learning-based models provided. Although significant progress has been made, more work is needed to fully accomplish the linking part of the problem, paying attention not only to the technical, but also to the biologically inspired measures. To help with this effort, the CTC will launch a new tracking-only benchmark, which will enable participants to tune and optimize their algorithms also to our biologically inspired measures. It is still, however, an open question how the biologically inspired tracking measures used in CTC can be reformulated as objective functions directly usable for optimization. It would also be useful to further investigate the impact of error propagation on tracking measures and on the accuracy of biological results in the definition of the track.

## Online content

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

[1]Centre for Biomedical Image Analysis, Faculty of Informatics, Masaryk University, Brno, Czech Republic. [2]IT4Innovations National Supercomputing Center, VSB – Technical University of Ostrava, Ostrava, Czech Republic. [3]Bioengineering Department, Universidad Carlos III de Madrid, Madrid, Spain. [4]Instituto de Investigación Sanitaria Gregorio Marañón, Madrid, Spain. [5]Optical Cell Biology, Instituto Gulbenkian de Ciência, Oeiras, Portugal. [6]Centro de Informatica, Universidade Federal de Pernambuco, Recife, Brazil. [7]Center for Advanced Methods in Biological Image Analysis, Beckman Institute, California Institute of Technology, Pasadena, CA, USA. [8]Division of Biology and Biological Engineering and Howard Hughes Medical Institute, California Institute of Technology, Pasadena, CA, USA. [9]Institute for Automation and Applied Informatics, Karlsruhe Institute of Technology, Eggenstein-Leopoldshafen, Germany. [10]The Elmore Family School of Electrical and Computer Engineering, Purdue University, West Lafayette, IN, USA. [11]Boston Children's Hospital and Harvard Medical School, Boston, MA, USA. [12]CIVA Lab, Department of Electrical Engineering and Computer Science, University of Missouri, Columbia, MO, USA. [13]Institut de Génomique Fonctionnelle de Lyon (IGFL), École Normale Supérieure de Lyon, Lyon, France. [14]Centre National de la Recherche Scientifique (CNRS), Paris, France. [15]Raysearch Laboratories AB, Stockholm, Sweden. [16]Department of Electrical and Computer Engineering, Drexel University, Philadelphia, PA, USA. [17]School of Electrical and Computer Engineering, Ben-Gurion University of the Negev, Beersheba, Israel. [18]Division of Medical Image Computing, German Cancer Research Center (DKFZ), Heidelberg, Germany. [19]Helmholtz Imaging, German Cancer Research Center (DKFZ), Heidelberg, Germany. [20]Interactive Machine Learning Group, German Cancer Research Center (DKFZ), Heidelberg, Germany. [21]Pattern Analysis and Learning Group, Department of Radiation Oncology, Heidelberg University Hospital, Heidelberg, Germany. [22]School of Computer Science and Engineering, University of New South Wales, Sydney, New South Wales, Australia. [23]Griffith University, Nathan, Queensland, Australia. [24]Biomedical Engineering Program and Ciberonc, Center for Applied Medical Research, Universidad de Navarra, Pamplona, Spain. [25]These authors contributed equally: Michal Kozubek, Carlos Ortiz-de-Solórzano. ✉e-mail: kozubek@fi.muni.cz; codesolorzano@unav.es

## Methods

### Gold standard reference annotations

The gold standard reference annotations (gold truth) of the real datasets used to evaluate the performance of the algorithms were created using majority voting over triplets of manual annotations. The segmentation gold truth consists of manually annotated cell instance masks for a subset of cells per video due to the immense manual effort that would be required for the segmenting of all cells in all frames. The detection and tracking gold truth consists of manually placed markers for all cells in the field of interest per video and their linking over time, respectively. The detection and tracking gold truth for the Fluo-N3DL-DRO, Fluo-N3DL-TRIC and Fluo-N3DL-TRIF datasets cover only a biologically relevant subset of cells at the beginning of imaging (early nervous system for *Drosophila melanogaster* and blastoderm of the beetle for *Tribolium castaneum*) and their lineages over time. These cells were annotated by one expert, with the lineages being carefully checked and curated by another expert.

### Silver standard reference annotations

The silver standard segmentation annotations were generated from the instance segmentation cell masks produced by the best-performing methods for particular datasets. To determine the best performers, we analyzed the results of all challenge participants who agreed to have their results used for further analysis. For each included dataset, we selected up to 16 best-performing methods that simultaneously achieved segmentation and detection scores above 50% of the reported human performance (see below for definition) on both the training and test datasets. Their results were merged using a modified version of the label fusion approach[38]. Specifically, the original approach iterates over individual markers in the detection gold truth, and collects, for each marker from the selected results, all segments that overlap the majority of this marker. Using pixel- or voxel-wise voting, the collected segments are fused into a single segment by taking only those pixels that appeared in more than two-thirds of the collected segments. The largest connected component of this segment is inserted into the output image. If there are fewer collected segments than the threshold of two-thirds, no output segment is created and inserted into the output image. However, it is possible that the insertion of another segment (resulting from the processing of another detection marker) into the output image may result in an attempt to overwrite the pixels or voxels of some previously inserted segment. Such intersections of pixels or voxels from two (or more) segments are detected, and the percentage of intersection pixels (PoIP) from the segment's original size is computed for every output segment. Output segments with PoIP > 10% are cleared entirely from the output image; otherwise, only their intersection pixels are cleared. The absence of output segments (false negatives), or reduction of their shapes, decreases the quality of the silver truth. Nevertheless, this approach still produces more reliable segmentation results than the standard SIMPLE and STAPLE algorithms, originally developed for combining single, larger objects in medical imaging.

A modification of this method was designed to improve both the quantity and quality of the final masks, specifically to increase detection accuracy (reducing false negatives) and improve segmentation accuracy (more accurate boundaries) compared with the gold standard. To achieve this, the threshold for the pixel- and voxel-wise voting was no longer fixed at two-thirds, but instead was optimized automatically for each video to maximize the segmentation score in relation to the gold standard. Additionally, if there were fewer input segments available for a given object than the threshold for a given video, the segment produced by the best-performing method was taken from the pool of selected available segments (if any). To address overlapping segments more effectively, the modified version uses a marker-based watershed method to find the boundary and to create touching segments from the overlapping ones. The markers are either the created

segments without the intersection pixels or voxels (PoIP ≤ 20%) or the corresponding detection markers (PoIP > 20%).

The silver standard segmentation annotations for all the real (not simulated) test datasets were generated using the submissions available before June 2022. These annotations were kept secret and were used only to compute dataset properties for this paper (see below). The silver standard segmentation annotations for the training datasets were generated using the submissions available by October 2020 to release the silver standard corpus in time for the ISBI 2021 competition. Due to the low number (less than 5) of submissions that met the quality threshold at that time, silver truth was not generated for the three largest embryogenic datasets (Fluo-N3DL-DRO, Fluo-N3DL-TRIC and Fluo-N3DL-TRIF) or for the newly introduced Fluo-C2DL-Huh7 dataset.

### Dataset properties

A set of properties introduced earlier was computed to characterize each dataset. These properties primarily include a set of measures calculated based on the silver standard segmentation annotations of test videos: signal-to-noise ratio, contrast ratio, internal signal heterogeneity of the cells (Het$_i$), heterogeneity of the signal between cells (Het$_b$), resolution (Res), regularity of the cell shape (Sha), cell spacing (Spa) measured as the average distance to the closest neighbor, an absolute change of the average intensity of the cells with time (Cha), and level of cell overlap in consecutive frames (Ove). The number of division events (Mit) was computed based on the gold standard tracking annotations. The remaining qualitative parameters were determined based on the visual inspection of the datasets: synchronization of division events (Syn), cells entering or leaving the field of view (Ent/Leav), and the presence of apoptotic cells (Apo) and of moving debris (Deb).

### Quantitative performance criteria

The primary measures used to evaluate the methods and create ranked leaderboards (Cell Segmentation Benchmark, CSB, and Cell Tracking Benchmark, CTB) are technical measures of interest mainly for the developers. The segmentation accuracy measure (SEG) evaluates the average intersection over union overlap (IoU) as a measurement of the overlap between the reference cell instance masks and the segmentation masks computed by an evaluated algorithm. The tracking accuracy measure (TRA) is a normalized weighted distance between the tracking solution computed by an evaluated algorithm and the reference tracking solution, with weights chosen to reflect the effort it takes a human curator to carry out the edits manually[39]. The TRA measure evaluates both detection and linking capabilities. Both SEG and TRA definitions can be found in Ulman et al.[9].

For CSB, a new detection accuracy measure (DET) has been introduced. It computes only the detection part of the TRA measure. The DET measure is very similar to the F3-Score, an extension of the F1-Score. While F1-Score is a harmonic mean of precision and recall, F3-Score favors recall over precision, which is advantageous in time-lapse imaging to prevent losing cells. Numerically, DET is defined as a normalized acyclic oriented graph matching (AOGM-D) measure for detection[39]:

$$DET = 1 - \min(\text{AOGM-D}, \text{AOGM-D}_0)/\text{AOGM-D}_0$$

where AOGM-D is the cost of transforming a set of nodes (representing cells) provided by the participant into the set of gold truth nodes, and AOGM-D$_0$ is the cost of creating the set of gold truth nodes from scratch (that is, it is AOGM-D for empty detection results). The minimum operator in the numerator prevents the final value from being negative in the case when it is cheaper to create the reference set of nodes from scratch than to transform the computed set of nodes into the reference one. The normalization ensures that DET always falls in the [0,1] interval, with higher values corresponding to better detection performance.

For ranking the algorithms within CTB or CSB, the overall performance (OP$_{CTB}$ or OP$_{CSB}$) is computed by averaging SEG and TRA or SEG

and DET values for each pair of test videos and averaging these averages, that is, $OP_{CTB} = 0.5 \times (SEG_{avg} + TRA_{avg})$ and $OP_{CSB} = 0.5 \times (SEG_{avg} + DET_{avg})$. All five measures thus take values in the interval [0, 1], with higher values corresponding to better performance.

To assess the algorithm performance also from a biologist's point of view, we have computed four additional measures introduced earlier[9], in answer to frequent biological questions. Complete tracks (CT) measures the fraction of reference cell tracks that a given method can reconstruct entirely from the frame in which they appear to the frame in which they disappear. CT is especially relevant when a perfect reconstruction of the cell lineages is required. Track fractions (TF) averages, for all detected tracks, the fraction of the longest continuously matching algorithm-generated tracklet with respect to the reference track. Intuitively, this can be interpreted as the fraction of an average cell's trajectory that an algorithm reconstructs correctly once the cell has been detected. Branching correctness (BC(i)) measures a method's efficiency at detecting division events with a tolerance of i frames. Finally, cell cycle accuracy (CCA) measures how accurate an algorithm is at correctly reconstructing the length of cell cycles (that is, the time between two consecutive divisions). Both BC(i) and CCA are informative about the ability of the algorithm to detect cell population growth (that is, proliferation). All of the biologically inspired measures take values in the interval [0,1], with higher values corresponding to better performance.

## Top-performing algorithms

CALT-US (*) is a segmentation-only algorithm that performs semantic segmentation with an optimized U-Net followed by post-processing probability maps to identify individual cells. It provides excellent results for 2D bright-field datasets with high signal heterogeneity (high-$Het_i$ and high-$Het_b$) but varying levels of Res and Spa (BF-C2DL-HSC, DIC-C2DH-HeLa and PhC-C2DL-U3T3), and for two 3D fluorescence datasets that, in general, have image quality dissimilar to 2D bright-field, as shown in Fig. 2 (Fluo-C3DL-MDA231 and Fluo-N3DH-CHO), indicating a high degree of versatility of the method, as evidenced by its top rank in the generalizability study (see 'Generalizability study' below). The algorithm uses a new regularization term for the loss that is designed to handle imbalanced classes and promote sharper segmentation, and introduces a novel four class-based semantic representation aimed to improve separation of cells, specially for those cases when small gaps exist between adjacent cells[40,41].

KIT-GE (3) uses a two-branched 2D U-Net segmentation architecture. One branch predicts cell distance maps in which each cell pixel represents a normalized distance to the nearest pixel not belonging to the same cell. The other branch predicts neighbor distance maps in which each cell pixel represents an inverse normalized distance to the nearest pixel of the nearest surrounding cell. Tracking is based on the estimation of inter-frame displacement using cross-correlation, followed by cell matching across frames using a coupled minimum cost flow function[42,43]. The method has been designed to segment dense (low Spa), convex cells (high Sha). Accordingly, good segmentation results are obtained for the low resolution (low-Res) BF-C2DL-HSC (Fig. 2). However, the combination of cell and neighbor distance maps with an adapted tracker for error correction provides reasonable results also for more complex cell shapes such as those in the low-Res bright-field datasets PhC-C2DL-PSC and BF-C2DL-MuSC, regardless of their, respectively, very high mitotic activity (high Mit) and high variability (low Ove), and in two completely different 3D fluorescent datasets (Fluo-C3DH-H157 and Fluo-C3DL-MDA321). This variety of modalities and image properties also demonstrates a high level of versatility.

DKFZ-GE is a segmentation-only method based on a self-configuring U-Net pipeline called nnU-Net. It performs well for densely annotated datasets (Fluo-N2DH-SIM+, Fluo-N3DH-SIM+, Fluo-C3DH-A549 and Fluo-C3DH-A549-SIM+). For each dataset, nnU-Net is trained from scratch using no additional external training data. Given that

nnU-Net is designed for semantic segmentation, instance segmentation was implemented through conversion into a two-class semantic segmentation problem (cell border and center), the predictions being converted back to cells by outgrowing the center predictions. The core strength of nnU-Net lies in the fact that its recipe for automatically configuring the segmentation pipeline was developed and optimized on the 10 datasets of the medical segmentation decathlon[44], yielding more robust design choices and hyperparameters than methods tuned on individual datasets.

KIT-GE (4) extends the concept of cell instance learning[45] by simultaneously segmenting and tracking cells using an efficient, real-time deep neural network (ERFNet[46]). This network predicts the offset of each cell pixel to its corresponding cell center in two consecutive frames. Because the ERFNet is small and has very few parameters, the model avoids overfitting on the training data. This method performs best on two low-Res, bright-field datasets (BF-C2DL-MuSC and BF-C2DL-HSC) and one 2D fluorescence dataset (Fluo-N2DH-SIM+).

KTH-SE (1) uses a Gaussian bandpass filter for noise removal. Segmentation is based on a seeded h-dome-based watershed transform, which separates clustered cell masks on a thresholded version of the filtered image. The tracking part of the algorithm uses a global track linking algorithm, which greedily adds tracks one at a time using the Viterbi algorithm. The track linking optimizes each track over all frames at once and can therefore make linking decisions based on prior and future frames. Because of that, it can handle spurious detections as well as under- and over-segmentation. This lightweight segmentation strategy has outstanding performance for most fluorescence datasets (Fluo-C2DL-MSC, Fluo-C3DH-H157, Fluo-N2DH-GOWT1, Fluo-N3DH-CE, Fluo-N3DH-CHO, Fluo-C3DH-A549-SIM+ and Fluo-N3DH-SIM+) of varying quality properties. It is also the most convenient and computationally efficient strategy for extremely large mesoscopic datasets. A closely related algorithm, KTH-SE (2) (ref. 36), top-1 performer for the embryonic datasets Fluo-N3DL-DRO and Fluo-N3DL-TRIC and top-3 for Fluo-N3DL-TRIF, takes advantage of the dynamic nature of the nuclei motion to preprocess the detected locations using a Gaussian mixture probability hypothesis density (GM-PHD) filter.

## Quality of annotations and human-level performance

To characterize the quality of gold standard segmentation and tracking annotations (gold truth, GT), we calculated the average and standard deviation performance of the three independent annotators relative to the gold truth that was established by merging the triplets of manual annotations. Submitting individual manual annotations as if they were standard submissions into the technical and biological measures, a set of multiple measurements is defined and denoted with the prefix 'M', as here, for example, for the SEG measure:

$$MSEG_{GT} = \{SEG_{Annotation\_k}, k = 1, 2, 3\}$$

where Annotation_k denotes the k-th manual annotation. Similarly, the sets $MDET_{GT}$, $MTRA_{GT}$, $MCT_{GT}$ and so on are obtained, from which their averages and standard deviations can be found in the quality control criteria sheets of Supplementary Data Tabs 5–9 (CSB), Tabs 10–14 (CTB) and Tabs 15–20 (CTB: biological measures), and are also summarized in Fig. 5. Note that low values of average human performance and high values of standard deviation are typically caused by the high level of difficulty of the dataset, not by poor work performance of the annotators. The variability of these human performance values can also serve as the baseline for algorithm performance comparisons. To this end, algorithms with scores in the range of the average manual scores ±1 s.d. can be considered to perform at the level of human annotators, and algorithms with scores above or below that range are said to perform better or worse, respectively, than the human annotators. Such a classification of the benchmarked submissions can be found in the Cell Segmentation Benchmark, Cell Tracking Benchmark and individual

sheets of Supplementary Data Tabs 5–9, Tabs 10–14 and Tabs 15–20, with the methods performing above and at the level of human performance being highlighted in green and blue, respectively.

To characterize the quality of silver standard segmentation annotations (silver truth, ST), we compared these annotations with the gold standard annotations (gold truth). Specifically, to evaluate how similar the masks in silver truth are to those in gold truth, we computed $SEG_{ST}$ using the standard SEG measure as if the silver truth was a participant's result. Similarly, the detection quality of the silver truth was calculated, and is denoted as $DET_{ST}$. Owing to how silver truth is constructed (see 'Silver standard reference annotations' above), $DET_{ST}$ can decrease from 1.0 only due to false negatives because silver truth annotations cannot have false positives. The $DET_{ST}$ values, therefore, implicitly reflect the coverage of the silver annotation. The $SEG_{ST}$ and $DET_{ST}$ values are summarized per dataset in Fig. 5.

Note the subtle differences in the interpretation of the quality measurements. Although all of the segmentation quality measurements are compared against the same reference, that is, the gold standard annotation for segmentation, $MSEG_{GT}$ provides merely feedback on the merger process by showing the similarities of the individual inputs to the product of their merger, thus reflecting the difficulty of the manual annotation task itself. $SEG_{ST}$, however, is compared directly with another (alternative) segmentation annotation. The $MDET_{GT}$ values are also unable to be compared directly with the $DET_{ST}$ values because the latter evaluate the annotations for which the creation process (unlike those in the former case) was supervised, as explained in the previous paragraph.

### Statistical analysis

The association between the quantitative image characteristics and segmentation (SEG) and tracking (TRA) performance in the Image Quality Versus Algorithm Performance section and between technical and biological measurements in the Correlation Between Technical and Biological Measures section was evaluated using the non-parametric Spearman's rank correlation, given that the normality of the data distribution was rejected in some cases. We define the levels of positive (negative for negative values of rho, respectively) correlation according to these intervals: from 0 to 0.2 as very weak or no association; from 0.2 to 0.4 as weak association; from 0.4 to 0.6 as moderate association; from 0.6 to 0.8 as strong association and from 0.8 to 1 as very strong association.

For a comparison of detection, segmentation and linking techniques in the Evolution of the Segmentation and Tracking Paradigms section, a non-parametric two-sided Kruskal–Wallis test with Dunn's post-hoc test was applied. We focused on a pairwise comparison of the techniques with machine learning techniques. The $P$ value of the post-hoc test was corrected for multiple comparison testing using Holm's correction. If only two groups were compared with each other, a non-parametric two-sided Mann–Whitney test was applied instead of the Kruskal–Wallis test. An absolute difference between the median values of two groups is expressed as 'diff'.

All boxplots used in the above mentioned sections (Image Quality versus Algorithm Performance, Correlation Between Technical and Biological Measures, and Evolution of the Segmentation and Tracking Paradigms) were constructed with a bold line representing the median value and a box showing an interquartile range. In the case of outliers, that is, values higher or lower than 1.5-fold the interquartile range, points (outliers) exceed the whiskers.

The Evolution of the Segmentation and Tracking Paradigms section includes plots with descriptive statistics for the machine learning and the non-machine learning techniques separately, by year. In these plots, median values are represented by a solid line; a dashed line plots the mean values. This analysis also consists of sums of the median and/or mean values per dataset over the years (Extended Data Figs. 3, 5 and 7). For each year, the median and/or mean value of the performance on

existing datasets was computed for each group (machine learning and non-machine learning) separately. These median and/or mean values were summed up and plotted in connected lines.

The significance level (alpha) was set at 0.05 in all of the statistical analyses. Furthermore, to screen out low-performing or poorly fine-tuned CTC submissions, only those algorithms that simultaneously achieved DET and SEG scores (in the case of the CSB submissions), and SEG and TRA scores (in the case of the CTB submissions) above 50% of the reported human performance were considered.

All statistical analyses were performed using R Statistical Software (v3.4.3; R Core Team)[47].

### Reporting summary

Further information on research design is available in the Nature Portfolio Reporting Summary linked to this article.

## Data availability

The training datasets with their reference annotations and test datasets used in the challenge are publicly available at the CTC website (http://celltrackingchallenge.net). With regard to the raw data for individual Figures and Tables, Source data are provided with this paper.

## Code availability

The evaluation routines used to produce the results reported in this article are freely available at the CTC website or the CellTrackingChallenge update site as a Fiji plugin, the source codes of which can be found at https://github.com/CellTrackingChallenge/. Furthermore, this public GitHub repository contains links to the executable versions of the individual algorithms and Colab Notebooks of those 11 participants who agreed to share their tools in a reusable form. The parameters used by the participants to produce their benchmarked results are listed on the CTC website.

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

## Acknowledgements

The authors thank J. Padilla Pérez, who worked for many hours on the annotation of the new datasets; J.-Y. Tinevez, who kindly added the CTC measures into the popular TrackMate software and who, with T. Pietzsch, developed the Mastodon software that became instrumental for us when preparing the tracking annotations of the large embryonic datasets; and the participants of the challenge not included in the list of authors of this analysis paper, listed on the challenge website (http://celltrackingchallenge.net/participants/). This work was funded by the Ministerio de Ciencia, Innovación y Universidades, Agencia Estatal de Investigación (MCIU/AEI/10.13039/50110011033) and FEDER funds UE under Grants RTI2018-094494-B-C22, TED2021-131300B-I00, PDI2021-122409OB-C22 (C.O.S.); Czech Ministry of Education, Youth and Sports national research infrastructure Czech-BioImaging projects LM2023050 and CZ.02.1.01/0.0/0.0/18_046/0016045 (M.M., V.U. and M.K.); Czech Science Foundation (GACR) grant GA21-20374S (P.M. and F.L.); European Regional Development Fund in the IT4Innovations national super-computing center–path to exascale project CZ.02.1.01/0.0/0.0/16_013/0001791 within the Operational Programme Research, Development and Education (V.U.); Czech Ministry of Education, Youth and Sports through the e-INFRA CZ project ID:90140 (V.U.); Ministerio de Ciencia, Innovación y Universidades, Agencia Estatal de Investigación Grant PID2019-109820RB-I00, MCIN/AEI/10.13039/501100011033/, co-financed by the European Regional Development Fund (ERDF), 'A way of making Europe' (P.D.-R., E.G.M., grant to A.M.-B.); BBVA Foundation under a 2017 Leonardo Grant for Researchers and Cultural Creators (A.M.-B.). NVIDIA Corporation for the donation of the Titan X (Pascal) GPU (P.D.-R., E.G.M., to A.M.-B.); the Gulbenkian Foundation, the European Molecular Biology Organization (EMBO) Installation Grant (EMBO-2020-IG-4734) (granted to the Optical Cell Biology laboratory at Instituto Gulbenkian de Ciência) and Postdoctoral Fellowship (EMBO ALTF 174-2022) (E.G.M.); Helmholtz Association program NACIP – Natural, Artificial and Cognitive Information Processing and Biointerfaces International Graduate School (BIF-IGS) (T.S. and R.M.), and HIDSS4Health – Helmholtz Information and Data Science School for Health (K.L. and R.M.); European Research Council, under the European Union Horizon 2020 programme, grant ERC-2015-AdG 694918 (K.S.); Helmholtz Imaging (F.I., P.F.J.); the Negev scholarship at Ben-Gurion University (A.A.); the Kreitman School of Advanced Graduate Studies (A.A.); Israel Ministry of Science, Technology and Space (MOST 3-14344 T.R.R.); the United States – Israel Binational Science Foundation (BSF 2019135 T.R.R.); Human Frontiers Science Program Grant RGP0043/2019 (A.R.C. and L.A.); the Brazilian funding agencies FACEPE, CAPES and CNPq (F.A.G.P., T.I.R.); Beckman Institute at Caltech (A.C., F.A.G.P.); Howard Hughes Medical Institute (E.M.M.); and the USA NIH NINDS R01NS110915 (K.P.) and USA ARL W911NF-18-20285 (K.P.).

## Author contributions

M.M. conceived and designed the analysis, collected the data, contributed data or analysis tools, performed the analysis and wrote the paper; V.U. conceived and designed the analysis, contributed data or analysis tools, and performed the analysis; P.D.-R. and E.G.M. collected the data and performed the analysis; T.N. performed the analysis; F.A.G.P., T.I.R., E.M.M., T.S., K.L., R.M., T.G., Y.W., J.P.A., R.B., N.M.A.-S., G.R., I.E.T., K.P., F.L., P.M., K.S., K.E.G.M., L.A., A.R.C., A.A., T.B.-H., T.R.R., F.I., P.F.J., K.H.M.-H. and Y.Z. contributed data or analysis tools; C.E. and A.U. collected the data, other contribution (annotated data from ground truth); E.M. and A.C. conceived and designed the analysis; and A.M.-B., M.K. and C.O.S. conceived and designed the analysis and wrote the paper.

## Competing interests

K.S. is employed part-time by LPIXEL Inc. All other authors have no competing interests.

## Additional information

**Extended data** are available for this paper at https://doi.org/10.1038/s41592-023-01879-y.

**Correspondence and requests for materials** should be addressed to Michal Kozubek or Carlos Ortiz-de-Solórzano.

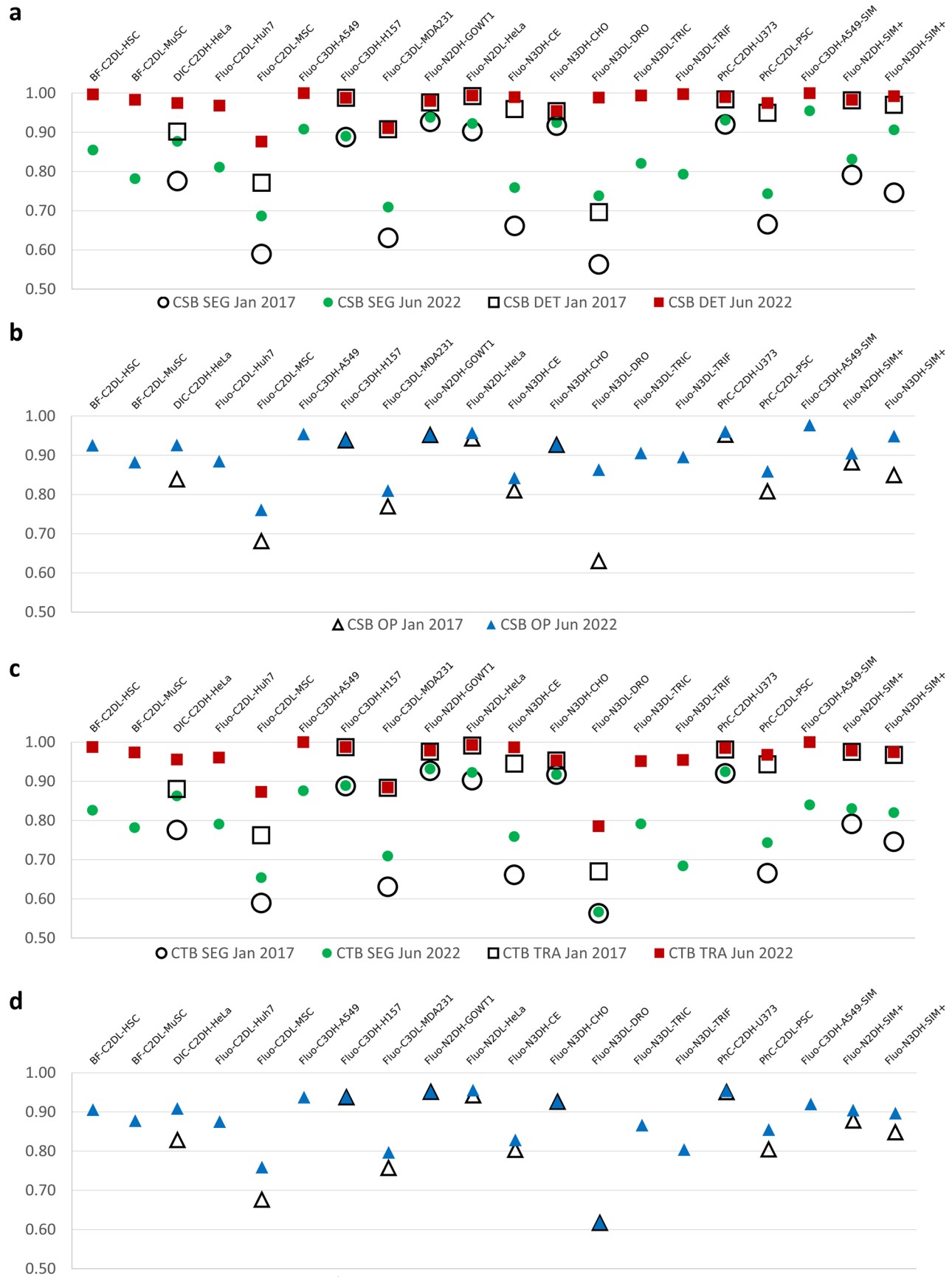

**Extended Data Fig. 1 | Evolution of technical measures from 2017 to 2022.**
(**a**) Evolution of CSB segmentation (SEG) and detection (DET) performance based on the top-ranked performing methods. (**b**) Evolution of CSB global (OP) performance based on the top-ranked performing methods. (**c**) Evolution of CTB segmentation (SEG) and tracking (TRA) performance based on the top-ranked performing methods. (**d**). Evolution of CTB global (OP) performance based on the top-ranked performing methods.

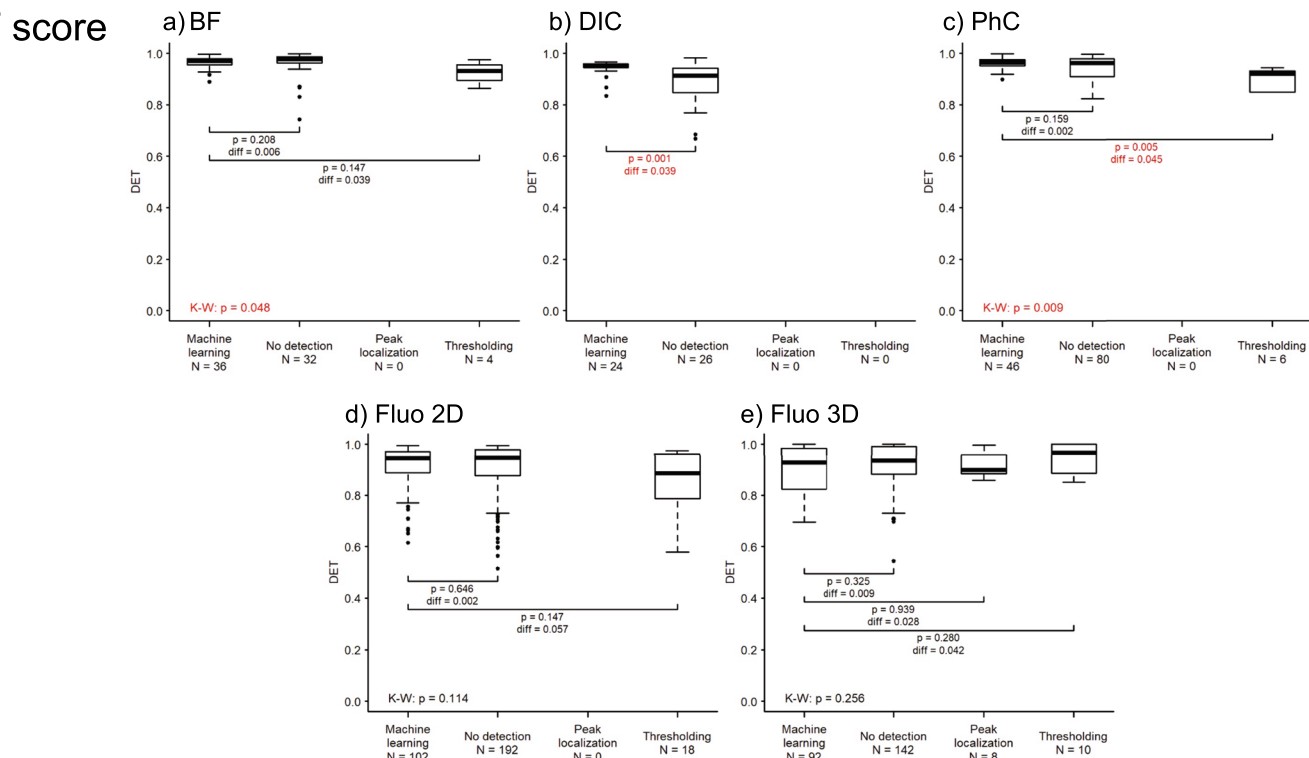

**DET score**

**Extended Data Fig. 2 | Detection (DET) performance according to detection technique and modality. a)** Bright-field (BF) **b)** DIC **c)** Phase Contrast (PhC), **d)** Fluorescence 2D (Fluo 2D) and **e)** Fluorescence 3D (Fluo 3D). For comparison of the techniques Kruskal–Wallis test with Dunn's post-hoc test were performed (or Mann–Whitney test in case of only two groups). The absolute difference between particular medians is reported as diff.

## Sum of DET score according to ML / non-ML techniques over years

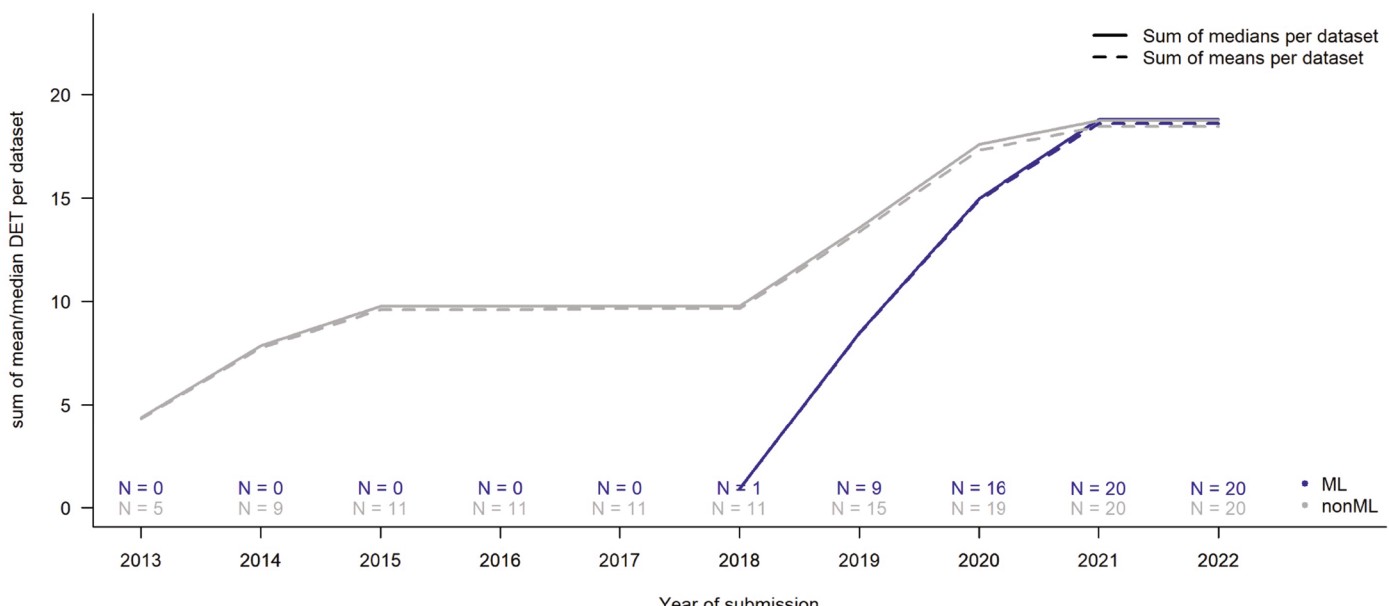

**Extended Data Fig. 3 | A sum of detection (DET) scores according to the use of machine learning (ML) or other (non-ML) techniques over years.** For each year, the median/mean value of the performance on existing datasets was computed for each group (ML and non-ML) separately. These median/mean values were summed up and plotted in connected lines.

# SEG score

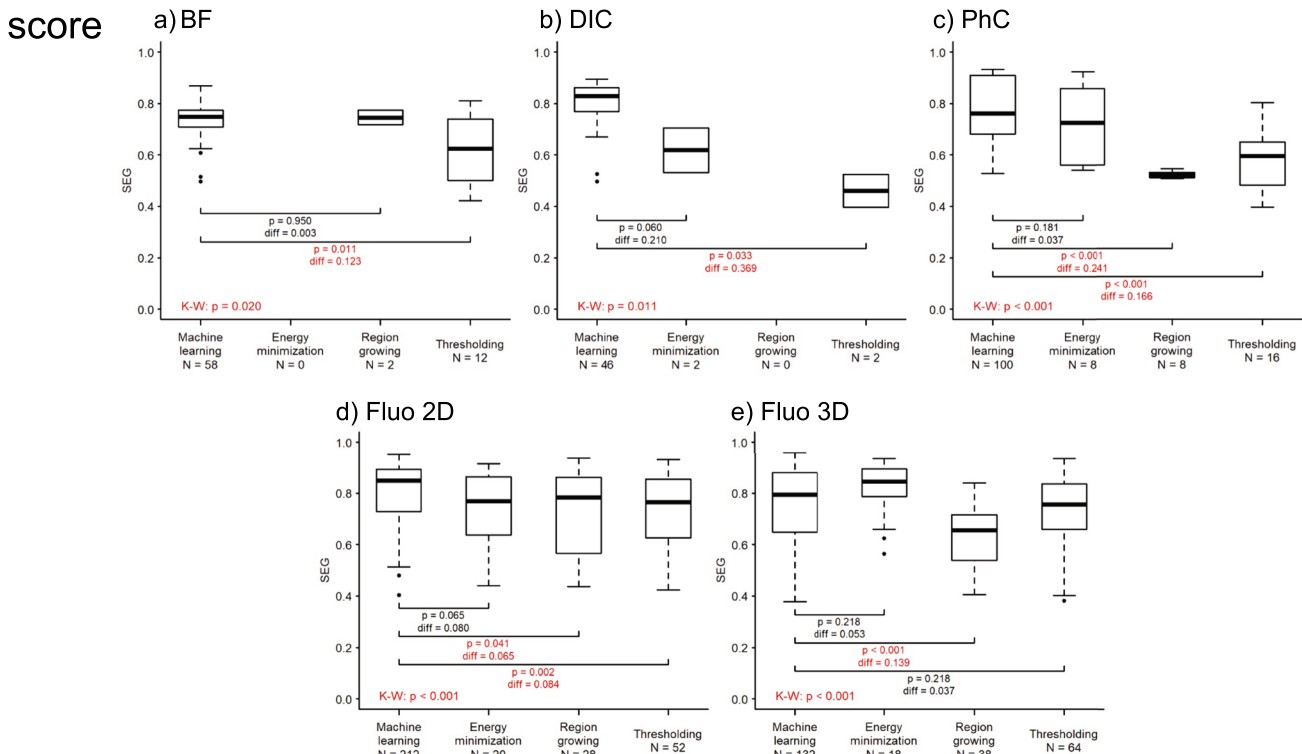

**Extended Data Fig. 4 | Segmentation (SEG) performance according to segmentation technique and modality. a)** Bright-field (BF) **b)** DIC **c)** Phase Contrast (PhC), **d)** Fluorescence 2D (Fluo 2D) and **e)** Fluorescence 3D (Fluo 3D). For comparison of the techniques Kruskal–Wallis test with Dunn's post-hoc test were performed. The absolute difference between particular medians is reported as diff.

## Sum of SEG score according to ML / non-ML techniques over years

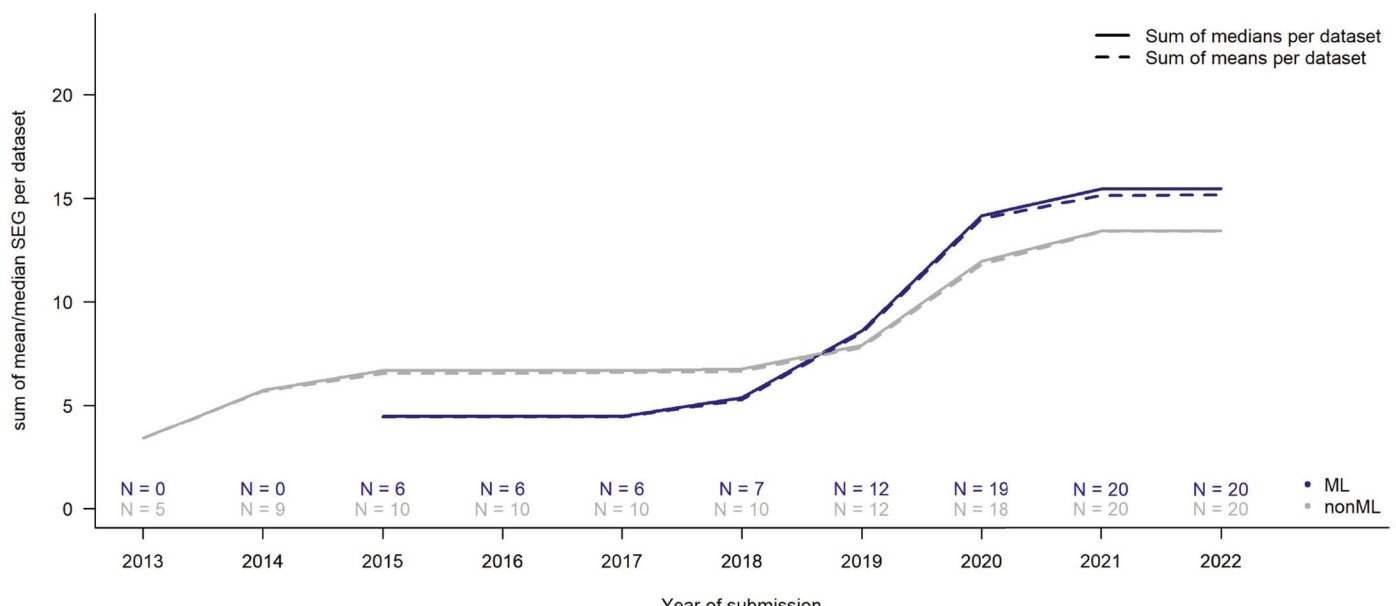

**Extended Data Fig. 5 | A sum of segmentation (SEG) scores according to the use of machine learning (ML) or other (non-ML) techniques over years.** For each year, the median/mean value of the performance on existing datasets was computed for each group (ML and non-ML) separately. These median/mean values were summed up and plotted in connected lines.

# TRA score

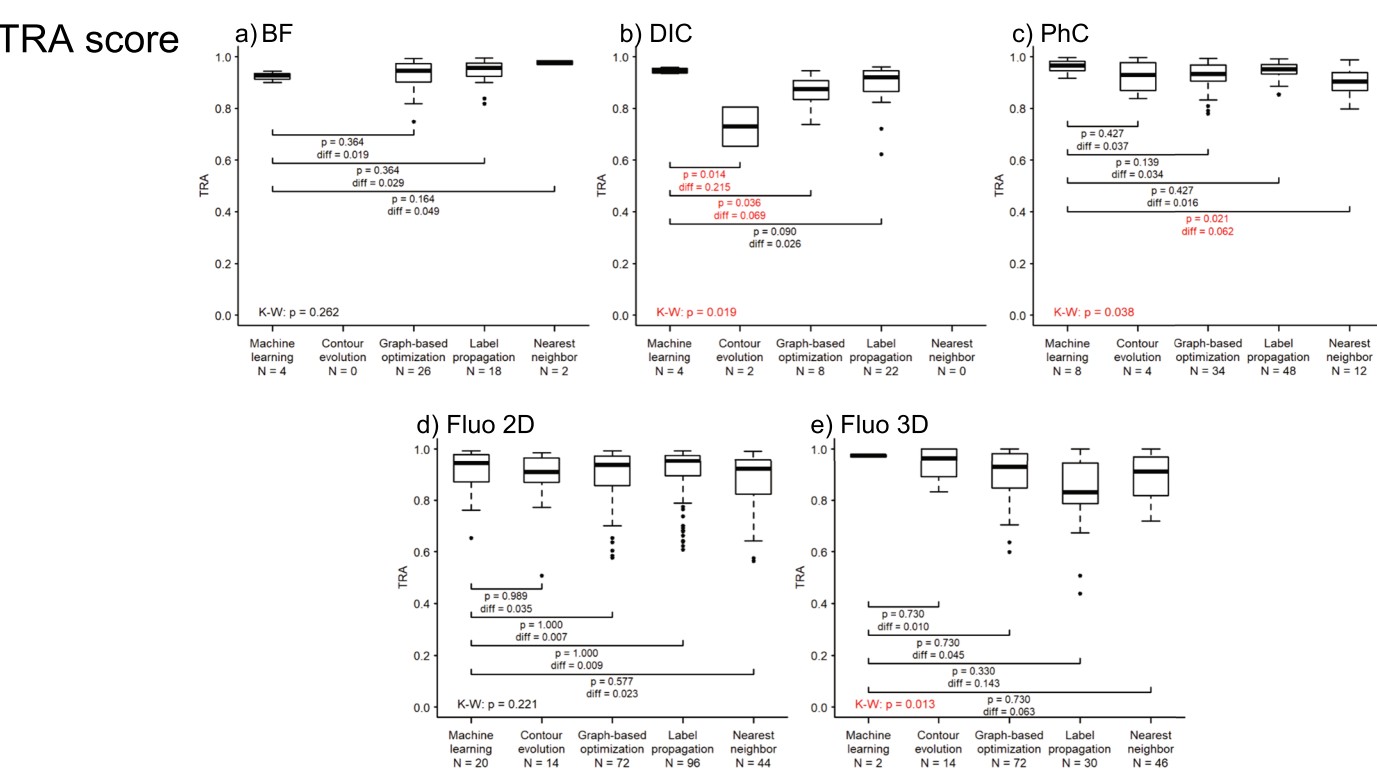

**Extended Data Fig. 6 | Tracking (TRA) performance according to linking technique and modality. a)** Bright-field (BF) **b)** DIC **c)** Phase Contrast (PhC), **d)** Fluorescence 2D (Fluo 2D) and **e)** Fluorescence 3D (Fluo 3D). For comparison of the techniques Kruskal–Wallis test with Dunn's post-hoc test were performed. The absolute difference between particular medians is reported as diff.

## Sum of TRA score according to ML / non-ML techniques over years

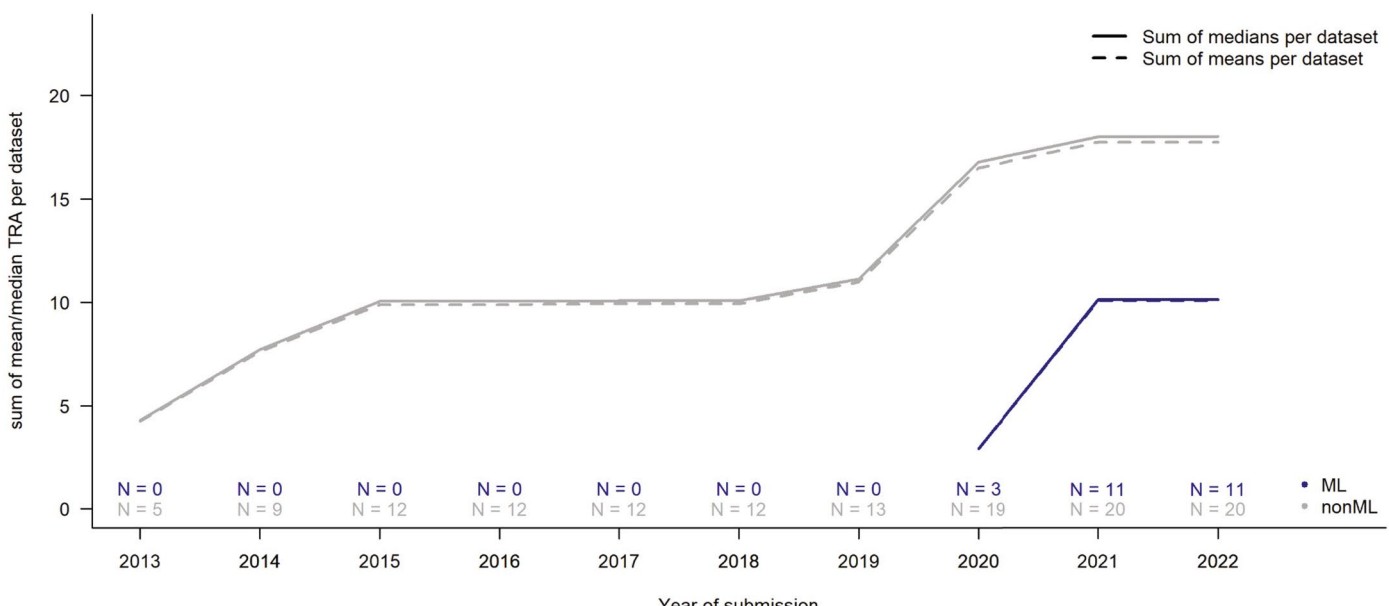

**Extended Data Fig. 7 | A sum of tracking (TRA) scores according to the use of machine learning (ML) or other (non-ML) techniques over years.** For each year, the median/mean value of the performance on existing datasets was computed for each group (ML and non-ML) separately. These median/mean values were summed up and plotted in connected lines.

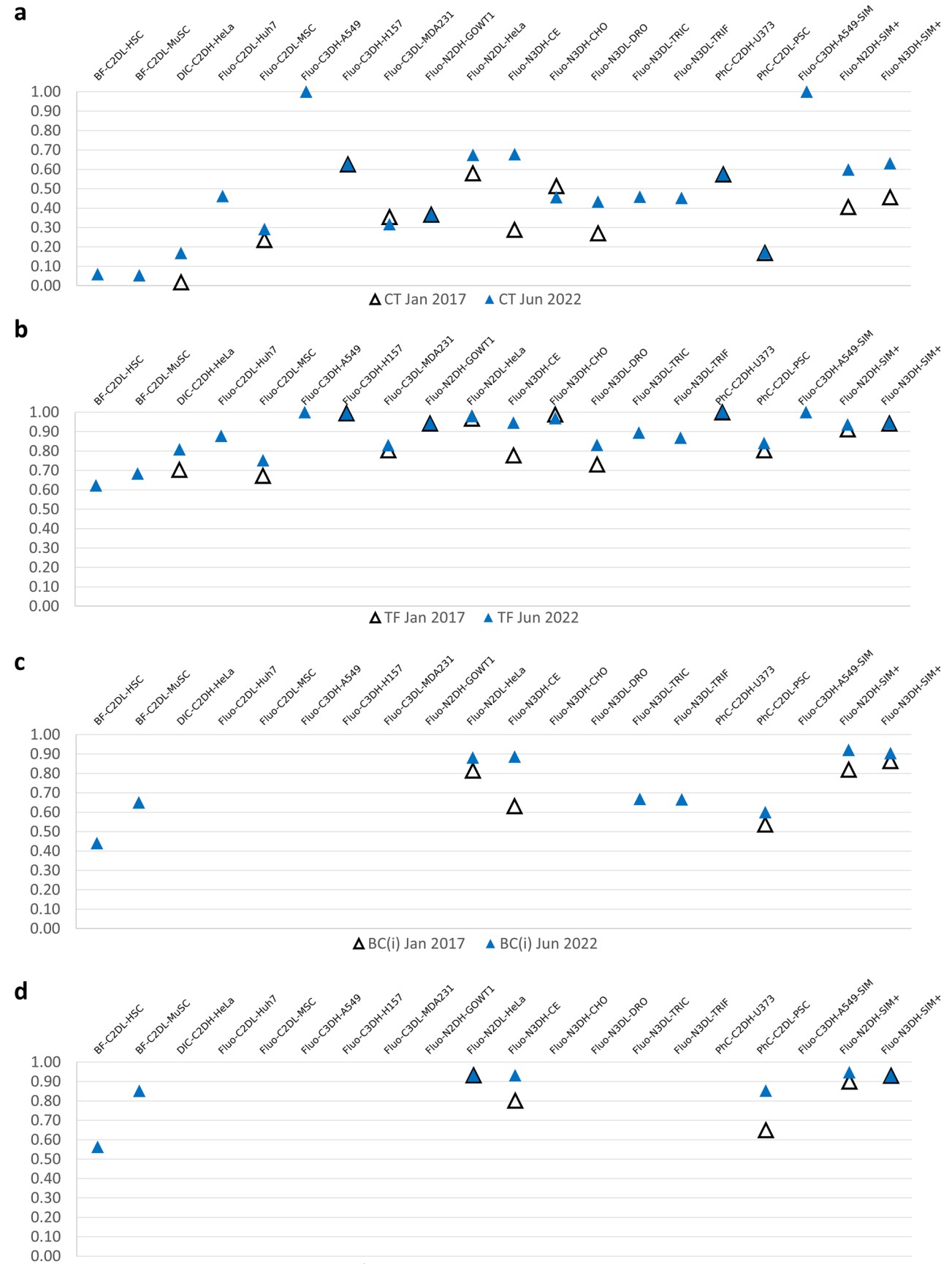

**Extended Data Fig. 8 | Evolution of biological measures from 2017 to 2022.**
(**a**) Evolution of complete tracks (CT) performance based on the top-ranked performing methods. (**b**) Evolution of track fractions (TF) based on the top-ranked performing methods. (**c**) Evolution of branching correctness (BC(i)) performance based on the top-ranked performing methods. (**d**) Evolution of cell cycle accuracy (CCA) performance based on the top-ranked performing methods.

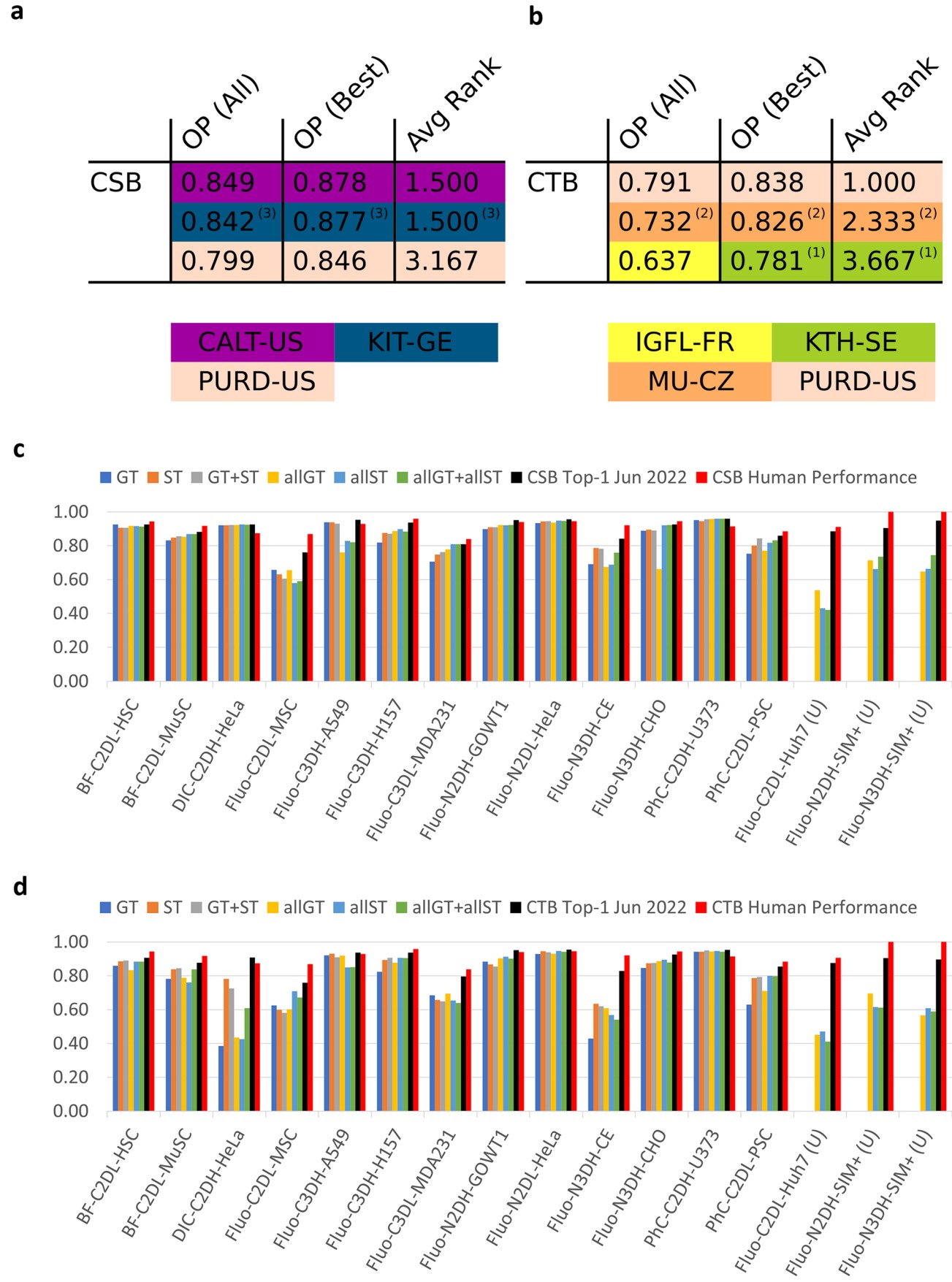

**Extended Data Fig. 9 | See next page for caption.**

**Extended Data Fig. 9 | Generalizability results.** (**a**) The top-3 most generalizable CSB methods. (**b**) The top-3 most generalizable CTB methods. (**c**) Results of the most generalizable CSB submission (CALT-US). (**d**) Results of the most generalizable CTB submission (PURD-US). The methods were ranked based on three different aspects: overall performance across all 78 evaluated cases, OP (All); overall performance of best training configurations per dataset, OP (Best); and average rank over training configurations, Avg Rank, leading to practically stable rankings across all the three ranking schemes used (see Supplementary Data (Tabs 21–33) (CSB) and (Tabs 34–42) (CTB) for more details). Dataset-specific algorithm configurations trained on gold truth (GT), silver truth (ST), or both (GT+ST) are compared to universal configurations trained on all 13 datasets (prefix all). The unseen data types are marked by (U) and naturally miss dataset-specific configuration results. **Legend for datasets names:** (Please see Fig. 2).

# Reporting Summary

Nature Research wishes to improve the reproducibility of the work that we publish. This form provides structure for consistency and transparency in reporting. For further information on Nature Research policies, see our Editorial Policies and the Editorial Policy Checklist.

## Statistics

For all statistical analyses, confirm that the following items are present in the figure legend, table legend, main text, or Methods section.

| n/a | Confirmed | |
|---|---|---|
| ☐ | ☒ | The exact sample size (*n*) for each experimental group/condition, given as a discrete number and unit of measurement |
| ☐ | ☒ | A statement on whether measurements were taken from distinct samples or whether the same sample was measured repeatedly |
| ☐ | ☒ | The statistical test(s) used AND whether they are one- or two-sided<br>*Only common tests should be described solely by name; describe more complex techniques in the Methods section.* |
| ☐ | ☒ | A description of all covariates tested |
| ☐ | ☒ | A description of any assumptions or corrections, such as tests of normality and adjustment for multiple comparisons |
| ☐ | ☒ | A full description of the statistical parameters including central tendency (e.g. means) or other basic estimates (e.g. regression coefficient) AND variation (e.g. standard deviation) or associated estimates of uncertainty (e.g. confidence intervals) |
| ☐ | ☒ | For null hypothesis testing, the test statistic (e.g. $F$, $t$, $r$) with confidence intervals, effect sizes, degrees of freedom and $P$ value noted<br>*Give P values as exact values whenever suitable.* |
| ☒ | ☐ | For Bayesian analysis, information on the choice of priors and Markov chain Monte Carlo settings |
| ☒ | ☐ | For hierarchical and complex designs, identification of the appropriate level for tests and full reporting of outcomes |
| ☐ | ☒ | Estimates of effect sizes (e.g. Cohen's *d*, Pearson's *r*), indicating how they were calculated |

*Our web collection on statistics for biologists contains articles on many of the points above.*

## Software and code

Policy information about availability of computer code

| Data collection | N/A No special software was used for data collection. |
|---|---|
| Data analysis | All the code used to produce the results reported in this article is freely available both at the CTC website and also as a Fiji plugin whose source codes can be found at https://github.com/CellTrackingChallenge/. Furthermore, this public github repository contains links to the executable versions of the individual algorithms and Colab Notebooks of those ten participants who agreed to share their tools. The parameters used by the participants to produce their submitted results are listed on the CTC website as well.<br>Analysis of the collected data was performed using R Statistical Software (v3.4.3). |

For manuscripts utilizing custom algorithms or software that are central to the research but not yet described in published literature, software must be made available to editors and reviewers. We strongly encourage code deposition in a community repository (e.g. GitHub). See the Nature Research guidelines for submitting code & software for further information.

## Data

Policy information about availability of data

All manuscripts must include a data availability statement. This statement should provide the following information, where applicable:
- Accession codes, unique identifiers, or web links for publicly available datasets
- A list of figures that have associated raw data
- A description of any restrictions on data availability

The training datasets with their reference annotations and test datasets used in the challenge are publicly available at the CTC website: http://celltrackingchallenge.net. As for the raw data for individual Figures and Tables:
Figure 1: Complete raw data available for download in the CTC website: http://celltrackingchallenge.net/datasets/. Source raw files used in the figure provided as Source Data Figures 1a, 1b, 1c, 1d, 1e, 1f, 1g

Figure 2: Calculated from raw test image data available for download on the CTC website: http://celltrackingchallenge.net/datasets/, from unreleased results of selected benchmarked algorithms, and from the secret tracking GT annotations, following the methodology described in the Dataset properties section in Online Methods and using the freely available Fiji plugins.

Figure 3: Compiled from the information contained in the CTC website ( http://celltrackingchallenge.net/participants/), summarized in Supplementary Data (Tabs 3-4).

Figure 4: Compiled from the data contained in Supplementary Data (Tabs 5-9) & (Tabs 10-14).

Figure 5: Calculated from the data obtained for Figure 2 and from secret manual GT annotations of test images, following the methodology described in the Section Quality of annotations and human level performance section in Online Methods and using the freely available Fiji plugins.

Figure 6: Compiled from the data contained in Supplementary Data (Tabs 15-20).

Extended Data Figure 1: Compiled from the data contained in Supplementary Data (Tabs 3-4), (Tabs 5-9) & (Tabs 10-14).

Extended Data Figure 2: Compiled from the data contained in Supplementary Data (Tabs 3-4) & (Tabs 5-9).

Extended Data Figure 3: Compiled from the data contained in Supplementary Data (Tabs 3-4) & (Tabs 5-9).

Extended Data Figure 4: Compiled from the data contained in Supplementary Data (Tabs 3-4) & (Tabs 5-9).

Extended Data Figure 5: Compiled from the data contained in Supplementary Data (Tabs 3-4) & (Tabs 5-9).

Extended Data Figure 6: Compiled from the data contained in Supplementary Data (Tabs 3-4) & (Tabs 10-14).

Extended Data Figure 7: Compiled from the data contained in Supplementary Data (Tabs 3-4) & (Tabs 10-14).

Extended Data Figure 8: Compiled from the data contained in Supplementary Data (Tabs 3-4) & (Tabs 15-20).

Extended Data Figure 9: Compiled from the data contained in Supplementary Data (Tabs 21-33) & (Tabs 34-42).

# Field-specific reporting

Please select the one below that is the best fit for your research. If you are not sure, read the appropriate sections before making your selection.

☒ Life sciences ☐ Behavioural & social sciences ☐ Ecological, evolutionary & environmental sciences

For a reference copy of the document with all sections, see nature.com/documents/nr-reporting-summary-flat.pdf

# Life sciences study design

All studies must disclose on these points even when the disclosure is negative.

| | |
|---|---|
| Sample size | N/A. This manuscript reports on procedures and software methods used to analyze images, not the actual use of those methods. There were no formal hypothesis either expected results in advance, therefore we collected the possible highest sample size regarding to the capacities. |
| Data exclusions | N/A (see comment about Sample size). Not to spoil the conclusions of the analyses by low-performing or poorly fine-tuned CTC submissions, only those algorithms that simultaneously achieved DET and SEG scores in case of the CSB submissions, and SEG and TRA scores in case of the CTB submissions, over halves of the reported human performance were considered. |
| Replication | N/A (see comment about Sample size) We did not perform any replication to have the groups independent as much as possible. |
| Randomization | N/A (see comment about Sample size) There was no randomization as we have no concerns about biased results due to chosen techniques. |
| Blinding | N/A (see comment about Sample size) There was no blinding as the participants chose the technique by their own to get the best results. |

# Reporting for specific materials, systems and methods

We require information from authors about some types of materials, experimental systems and methods used in many studies. Here, indicate whether each material, system or method listed is relevant to your study. If you are not sure if a list item applies to your research, read the appropriate section before selecting a response.

## Materials & experimental systems

| n/a | Involved in the study |
|---|---|
| ☒ ☐ | Antibodies |
| ☒ ☐ | Eukaryotic cell lines |
| ☒ ☐ | Palaeontology and archaeology |
| ☒ ☐ | Animals and other organisms |
| ☒ ☐ | Human research participants |
| ☒ ☐ | Clinical data |
| ☒ ☐ | Dual use research of concern |

## Methods

| n/a | Involved in the study |
|---|---|
| ☒ ☐ | ChIP-seq |
| ☒ ☐ | Flow cytometry |
| ☒ ☐ | MRI-based neuroimaging |

