## [Peer Review File · Nature Methods]

Peer Review Information

Manuscript Title: The Cell Tracking Challenge: 10 years of objective benchmarking

Corresponding author name(s): Carlos Ortiz de Solórzano

Editorial Notes: n/a

Reviewer Comments & Decisions:

Decision Letter, initial version:
--

Dear Carlos,

Your Analysis, "The Cell Tracking Challenge: 10 years of objective benchmarking", has now been seen by three reviewers. As you will see from their comments below, although the reviewers find your work of considerable potential interest, they have raised a number of concerns. We are interested in the possibility of publishing your paper in Nature Methods, but would like to consider your response to these concerns before we reach a final decision on publication.

We therefore invite you to revise your manuscript to address these concerns. We ask that you address the requests for additional quantitative analyses made by reviewer 1 while updating the manuscript to reflect the questions and concerns of all reviewers. We also ask that in your discussion you envision the future of the challenge, and whether it will embrace other modalities or other types of tracking challenges, as mentioned by the reviewers.

* include a point-by-point response to the reviewers and to any editorial suggestions

* please underline/highlight any additions to the text or areas with other significant changes to facilitate review of the revised manuscript

* address the points listed described below to conform to our open science requirements

* ensure it complies with our general format requirements as set out in our guide to authors at www.nature.com/naturemethods

* resubmit all the necessary files electronically by using the link below to access your home page

[Redacted] This URL links to your confidential home page and associated information about manuscripts you may have submitted, or that you are reviewing for us. If you wish to forward this email to co-authors, please delete the link to your homepage.

We hope to receive your revised paper within 2-3 months. If you cannot send it within this time, please let us know. In this event, we will still be happy to reconsider your paper at a later date so long as nothing similar has been accepted for publication at Nature Methods or published elsewhere.

OPEN SCIENCE REQUIREMENTS

REPORTING SUMMARY AND EDITORIAL POLICY CHECKLISTS

Please note that these forms are dynamic ‘smart pdfs’ and must therefore be downloaded and completed in Adobe Reader. We will then flatten them for ease of use by the reviewers. If you would like to reference the guidance text as you complete the template, please access these flattened versions at <http://www.nature.com/authors/policies/availability.html>.

DATA AVAILABILITY

All novel DNA and RNA sequencing data, protein sequences, genetic polymorphisms, linked genotype and phenotype data, gene expression data, macromolecular structures, and proteomics data must be deposited in a publicly accessible database, and accession codes and associated hyperlinks must be provided in the “Data Availability” section.

Please include a “Data availability” subsection in the Online Methods. This section should inform readers about the availability of the data used to support the conclusions of your study, including accession codes to public repositories, references to source data that may be published alongside the paper, unique identifiers such as URLs to data repository entries, or data set DOIs, and any other statement about data availability. At a minimum, you should include the following statement: “The data that support the findings of this study are available from the corresponding author upon request”, describing which data is available upon request and mentioning any restrictions on availability. If DOIs are provided, please include these in the Reference list (authors, title, publisher (repository name), identifier, year). For more guidance on how to write this section please see: <http://www.nature.com/authors/policies/data/data-availability-statements-data-citations.pdf>

CODE AVAILABILITY

Please include a “Code Availability” subsection in the Online Methods which details how your custom code is made available. Only in rare cases (where code is not central to the main conclusions of the paper) is the statement “available upon request” allowed (and reasons should be specified).

MATERIALS AVAILABILITY

ORCID

Nature Methods is committed to improving transparency in authorship. As part of our efforts in this direction, we are now requesting that all authors identified as ‘corresponding author’ on published papers create and link their Open Researcher and Contributor Identifier (ORCID) with their account on the Manuscript Tracking System (MTS), prior to acceptance. This applies to primary research papers only. ORCID helps the scientific community achieve unambiguous attribution of all scholarly contributions. You can create and link your ORCID from the home page of the MTS by clicking on ‘Modify my Springer Nature account’. For more information please visit www.springernature.com/orcid.

Sincerely,
Rita

Rita Strack, Ph.D.
Senior Editor
Nature Methods

Reviewers' Comments:

Reviewer #1:

Remarks to the Author:

The manuscript by Maska and colleagues 'The cell tracking challenge: ...' provides an update on the CTC, a community benchmarking effort for cell segmentation and tracking algorithms for 2D and 3D time-lapse microscopy analysis. These algorithms are of central importance for many qualitative and quantitative investigations in cell and developmental biology, and others. Compared to the initial publication on the CTC in Nature Methods (ref. 9), the manuscript describes important new developments in the initiative and in the field in general. These concern specifically (i) increased numbers of image data sets and algorithm evaluations in the CTC, including developments over time; (ii) extended scope of annotations based on predictions by highest-performing algorithms (silver truth, ST), compared to human annotations (gold truth, GT); (iii) very valuable analyses of generalizability of algorithms, which are important for (existing and new) machine-learning (ML) based approaches; and (iv) a general shift (e.g., as indicated by numbers of new submissions) towards ML-based segmentation and tracking.

The combination of the above factors makes the manuscript an important and timely contribution to capture the state of the art as well as recent developments in biological image analysis. Due to the setup of the CTC, it relies on objective evaluations of methods using quantitative criteria, having the potential for an unbiased review that can address two (divergent) types of questions the readership may be interested in, namely (i) which segmentation and tracking methods to use or to further develop to tackle specific biological problems, and (ii) what the state of the field is more globally, and to what extent recent developments such as ML-based methods have contributed to improving it.

The manuscript aims to address both types of questions, but it appears more successful in addressing detailed questions of type (i) than more global questions of type (ii). A revision should, in particular, aim to leverage the wealth of data generated during the past years of the CTC by a more extended analysis. Important aspects of comparisons of human and machine performance and their relations to GT annotations need clarifications. Finally, the manuscript is less self-contained (in the sense of being accessible to a non-specialist reader) than ref. 9. Detailed suggestions for addressing these aspects are given below.

Major comments:

(i) Data analysis: The various iterations of the CTC as well as the expanding datasets have created a large dataset that is only explored in a qualitative manner. More formal analyses using simple statistical methods (e.g., ordinal or logistic regression) could exploit this data and help address the following questions more quantitatively; they seem important for the past and future development of the field (and are partially already discussed in the manuscript):

(i.a) Are dataset qualities and algorithmic performances (data in Table 1 and raw data for all algorithms) correlated, and if so, which features of datasets determine the difficulty of segmentation and tracking tasks? Answers could both help biologists gauge the success rate of analyzing a particular type of experiment, and developers focus future efforts.

(i.b) Are specific techniques (classification in Fig. 2b) related to algorithmic performance? Analysis results could support the conclusion on ML-based methods replacing traditional methods more quantitatively.

(i.c) Do trade-offs exist between technical (Table 1, Figure 3) and biological (Figure 4) performance and what determines biological performance of algorithms? Biological performance substantially lags technical performance, and it will be important for future developments of methods and / or the CTC itself to delineate (and discuss) avenues for improving biological performance, e.g. by expanding on p.10, 'However, as the methods are not directly optimized ...'.

(i.d) Do technical and biological performance evolve over time incrementally with existing methods, or by introducing new types of (ML) methods? The Figs. S1 and S2 give indications of performance improvements per dataset, but not in this more detailed sense.

(ii) Human vs machine performance: Definitions such as p.5 state that reference annotation quality is 'measured as inter-annotator variability for GT'. It is unclear how this variability was determined (e.g., if based on binary decision problems such as existence of a link in tracking, for three annotators values

such as those reported for TRA_hum in Table 1 are not possible). Importantly, the presentations of human performance in Table 1 and Fig. 5) suggest comparability of human and algorithmic measures, which by the definitions (measures of variance vs measures of overlap with or transformability to human-derived GT) are not plausible. Please clarify definitions of human measures and clearly indicate / discuss comparability or non-comparability of measures in the text and captions.

(iii) Accessibility: In contrast to ref. 9, the present manuscript does not introduce much of the field's terminology, or of the technical approaches, to a non-specialist reader. It may not be feasible or desirable to expand the manuscript accordingly, but I suggest careful revision to avoid unexplained technical terms as far as possible. For example, a sentence such as (p. 9/10) 'The main contribution of this work consists of a new regularization term for the Cross-Entropy loss function based on Youden's statistic, specifically designed to cope with highly imbalanced classes and weak annotations¹³.' will be impenetrable for most readers of Nature Methods. A revision may consider more high-level descriptions, shortening the discussion of individual algorithms, or moving method details to SI text.

Minor comments:

(i) Abstract: 'These improvements include the creation of a new segmentation-only benchmark ...'. Given the lower performance of tracking compared to segmentation, do you consider a similar tracking-only benchmark for future CTC iterations?

(ii) p.2, last para, 'Notably, the state-of-the-art U-Net architecture was among the top-performing approaches for cell segmentation¹⁰.': ref. 10 supports U-Net, not its evaluation in the CTC (ref. 9). In addition U-Net architectures were only in the top performers for selected (contrast-enhanced) data sets, according to ref. 9.

(iii) p.3, 'mesoscopic-size videos' – please define term.

(iv) Table 1 and general: Avoiding abbreviations in annotations when possible (e.g., column labels) and adding structuring annotations (e.g., imaging mode or cell type for rows, dataset properties vs performances) could help the reader comprehend the complex data presentations more easily.

(v) p.5, ST and GT: Please give indications of the quality of the ST annotations compared to GT annotations (for the subset of cells / tracks with overlap between the two), which is important to evaluate results of the generalizability study regarding training on GT, ST, and GT+ST.

(vi) Fig. 2a: It would be informative to add the number of submissions for the different tasks, to give an indication of the problems the field is focusing on.

(vii) p.10, generalizability study: Please explain OP_CSB at first use.

(viii) p.10, 'The "comparable score" criterion was defined as differing at most by the standard deviation of the human annotations for a particular dataset.' Definition is unclear, please see major comment (ii) above.

(ix) p.11, selection of unseen data types: Please discuss (ideally with the measures in Table 1) to what extent / in which aspects these 'data types' (rather: datasets) differed from the training data.

(x) p.13, 'an easy way to easily ... easily.' – Please revise.

(xi) p. 16, last para of discussion: Please comment on biological relevance / open tasks as well (see major comment i.c).

(xii) Data availability: To strengthen reproducibility of the present analysis, providing a static dataset in a separate repository (instead of links to the CTC challenge web site, which will be changing) seems mandatory, specifically for raw data in figures and tables, and for method parameters.

Reviewer #2:

Remarks to the Author:

GENERAL COMMENTS

Cell tracking challenging (CTC) is a competition aiming for boosting cell tracking techniques in 2D and 3D time-lapse images. This report has described the new data, performance criteria, algorithms, and test results for CTC since the last report in 2017. In comparison to 2017, more datasets for training and testing, including those of large sizes, were added. A new criterion focused solely on cell detection was added. The accuracies of these new submitted algorithms in segmentation and tracking on various test datasets were compared. The top-ranked algorithms with high accuracies across many datasets were discussed in terms of performance and techniques. Some of the top-ranked algorithm's generalizability was evaluated on both images with seen datatype and images with the more challenging unseen datatype. The re-usability of some of the algorithms have also been discussed.

This report revealed the advancements in cell tracking techniques and the recent trends in cell tracking methods related to the deep learning techniques. The descriptions of the top-ranked algorithms and their generalizability/re-usability will assist software developers and biologists users in selecting appropriate tracking methods to analyze the growing number of bio-medical images. I believe these new analyses are exciting and should be of great interest to a wide range of readers. On the other hand, I recommend that authors conduct additional analysis to assist readers in more easily selecting the appropriate techniques.

MAYOR COMMENTS FOR THE ALGORITHMS ANALYSIS:

The current manuscript lacks an analysis of the relationships between image properties and proper tracking techniques.

One of the most critical implications of this paper is that it can help developers and biologists choose appropriate methods for tracking cell of specific image datatype. Given that the authors have listed the dataset properties (Table 1) and classified the segmentation/tracking techniques used in each submitted algorithm (Fig. 2B), it would be much more interesting if the authors could further analyze the relationships between the dataset properties and the appropriate techniques.

For example, the results show that no method can achieve the best performance across all datasets (Fig. 3 and 5). This indicates that some techniques must be more appropriate for certain image properties, while others may be more appropriate for different image properties. Such relationships should be analyzed. Furthermore, if any image properties remain a significant obstacle to all methods, they should be described.

Fig. 2B shows that some techniques, such as U-Net for detection/segmentation and Overlap/Distance based techniques for tracking (linking), are more popular than others. Do these popular techniques produce more accurate results on some or all datasets? Are there any techniques that are less widely used but produce better results in some or all datasets?

A more general question that covers questions above will be: in what scenarios/image properties each technique excel? Although the authors have mentioned the use of specific techniques when describing the top-ranked algorithms, a more systematic analysis will provide readers with more solid information for selecting appropriate segmentation/tracking techniques.

MINOR COMMENTS FOR THE ALGORITHMS ANALYSIS:

In the discussion, authors emphasized the big change since 2017 is the use of deep learning techniques, especially in cell detection/segmentation. The data in Fig. 2B is consistent with this conclusion. The question is: are there any big advancement in tracking (linking) techniques, which should also be analyzed and described. If not, is it because the tracking algorithm is already mature and do not need large improvements? Or is it because the tracking (linking) part is relatively easier on the datasets used in this competition and do not need more advanced techniques?

MINOR COMMENTS FOR THE DATASETS:

1. New datasets have been added to CTC since 2017. I agree that increasing the number of datasets helps to better test the algorithms' performance in a variety of scenarios. However, the authors should explain the reason and the results of adding these data. For example, are these new datasets biologically important, and if so, why? Is it difficult to track some of the new datasets using previous methods? If so, why were they previously difficult to track, and why can new methods now track them?
2. Despite the name cell tracking challenge, the datasets used by CTC are essentially limited to cells in culture or embryos, according to the descriptions on the CTC website. Although these types of cell

images have been studied for many years and have been the primary target for developing tracking methods, they do not represent all cell image types. New methods for tracking cells in mature organs such as brains and hearts in alive animals [1, 2], which have fewer/no divisions but much larger rotational/deformational motions than cells in culture/embryos, have emerged in recent years. The authors should make it clear to the readers whether the goal of CTC is to track cells during development or to run a competition on any type of time-lapse cell image. If the latter is the goal, it may be more appropriate to state unequivocally that the dataset used in the current competition is limited and should be expanded in the future.

REFERENCES:

- [1] Nguyen, J. P., Linder, A. N., Plummer, G. S., Shaevitz, J. W. & Leifer, A. M. Automatically tracking neurons in a moving and deforming brain. *PLoS Computational Biology* 13, (2017).
- [2] Wen, C. et al. 3DeeCellTracker, a deep learning-based pipeline for segmenting and tracking cells in 3D time lapse images. *eLife* 10, (2021).

Reviewer #3:

Remarks to the Author:

Thanks to the development of microscopic techniques, there is a need to quantify image phenomena, which is logically preceded by the need to segment the image parts - cells. The paper is a logical follow-up to a similar report published five years ago, when a group of authors summarized the results of segmentation pipelines on realistic and often difficult datasets. The current update of this report reflects the progression in the field over the last 5 years, in particular the dominance of deep learning algorithms (and the pitfalls associated with them) and the proliferation of new microscopy techniques.

The current publication substantially follows the proven and rigid architecture and methodology of the 2017 Cell tracking challenge paper the history of the Challenge. The paper is mostly well understandable and because of the mentioned continuity, it is largely pointless to contradict the analytical procedures defined earlier. From a biologist's point of view, I am also very positive about the use of biological metrics for tracking. Now, in the current year, it is more about the urgent need to summarize the advances done in last 5 years. This review is therefore more concerned with the clarity and readability of the text.

The article reports evident progress in image segmentation. It is concluded that at least some aspects of tracking/segmentation are considered solved. Also, big improvement is observed on datasets considered difficult-to-track back in 2017. The CTC team increased the spectrum of datasets, so it better covers the spectrum of currently used methods, also including extremely challenging embryonal tomograms. The emphasis of this year's summary is mostly associated with deep-learning-specific issues, that is

generalizability and problems with manual annotation. The paper is well written, but, in particular, the “new” sections would benefit corrections/clarifications.

Interestingly, for the generalizability, the authors report that the improvement obtained when increasing the complexity of the training data is small, but on the other hand, still, the performance on unseen dataset is dramatically lower. This is an interesting observation in the perspective of the fact that two of these three datasets are simulated (N2DH-SIM+ and N3DH-SIM+). Can this fact play a role? These simulated datasets are similar to some of the real ones in the CTC (therefore, why the scores are not higher?). At the same time, the simulated data potentially degrades the results by introducing artificially generated images. Could the authors comment that?

In the generalizability section the assessment on the unseen dataset is of great interest to readers, however, some sentences are misleading, could the authors please choose different words or clarify the following: “Regarding the performance of these methods,... we can appreciate a similar behavior, where the scores obtained stay far from the best-performing methods...”

In the end of the introduction, also the aspect of the network interpretability is mentioned, however, this is not discussed in the next sections of the article.

Could the authors please comment the segmentation and tracking performance of simulated datasets (also in relation to the note on the SIM dataset usage in the generalizability section)? Specifically, the OP for both segmentation and tracking is slightly lower compared to somehow comparable N2DH and N3DH datasets. Is there any chance the lower OP is given because the simulated nature of the datasets, or is this effect purely random?

One of the aspects of the CTC advances is the creation of a silver truth based on the intelligent combination of the best-submitted results; the procedure is undestandably explained in the „Reference annotations“ section, however, in the first mentions, that is abstract, introduction and Table 1 legend its description is difficult to understand and the reader needs to jump between the online parts of the article. A short explanation in the introduction or the choice of other words might clarify how the silver truth was created and explain the term silver standard with better words.

Last three columns of the table 1, following the previous comment on the definition of the silver standard, it is difficult to imagine what is being compared against what (gold vs silver standard). Please consider creating a schematic illustration or adding text explanation.

There is some inconsistency between Table 1 and the subsection "reference annotations" in the names used: inter-annotator variability, match with GT for ST and SEGhum, SEGfus, TRAhum in the table. Unification would be beneficial to understand

The datasets used for segmentation-only task are the same as those used for tracking, that is, a timelapse of consequent frames which share a lot of visual similarities between each other. Then, if two subsequent frames are then used „one-for-training and one-for-validation“ (are they?), algorithms can achieve falsely better results. Can the authors how the frames for a segmentation task were handled, how (or if) this phenomenon exists and/or matters?

The section „Analysis of the technical performance of the submitted algorithms“ describes the architecture of the top-performing methods. As a biologist, this is outside the scope of my expertise.

The last decade has witnessed a significant expansion of native cell microscopy techniques in biology beyond fluorescence or intensity. Properties of light - polarization, phase, scattering, or spectroscopic techniques like Raman or Brillouin microscopy and many others are being used by biologists. While not all of them achieve the robustness of datasets necessary for learning networks, the need to track/segment exists for them as well. Therefore, another challenge for the future is to work with very small datasets or with more "exotic" techniques. For generalizability, these might be even bigger challenges than learning simulated fluorescence datasets. How can this be reflected in future challenges? Please consider this more as a comment, I understand that the subject of the manuscript is different.

Minor comments:

Table 1, description of the columns would benefit in mentioning the abbreviation prior each explanation, e.g. "SNR, Signal-to-noise-ratio; CR, contrast ratio,..."

Fig. 3 the abbreviation "OP" for overall performance should be explained in text or figure legend. In the current form the abbreviation is explained just in supplementa.

Fig. 1d, Fluo-C3DH-A549, representativeness of the figure: are majority of cells isolated like shown in the representative fig., or is a proportion of cells touching?

Supplementary figures 1 and 2 please consider adding the dataset legend to the x axis of the chart instead of mentioning in the figure legend.

In the section "Analysis of the technical performance of the submitted algorithms": please explain the meaning of "the CSB highlighted datasets"

Supplementary methodology, Quality of annotations and human level performance, it is mentioned that the quality control criteria is present also in supplementary table with BIO measures. However, compared to SCB and CTB excel no such list in BIO excel is present.

Section generalizability, something is wrong with the sentence „It also highlights the importance that those methods be provided with clear training instructions to make them usable for new types of data.“

Author Rebuttal to Initial comments

“Reviewer #1:

Remarks to the Author:

The manuscript by Maska and colleagues ‘The cell tracking challenge: ...’ provides an update on the CTC, a community benchmarking effort for cell segmentation and tracking algorithms for 2D and 3D time-lapse microscopy analysis. These algorithms are of central importance for many qualitative and quantitative investigations in cell and developmental biology, and others. Compared to the initial publication on the CTC in Nature Methods (ref. 9), the manuscript describes important new developments in the initiative and in the field in general. These concern specifically (i) increased numbers of image data sets and algorithm evaluations in the CTC, including developments over time; (ii) extended scope of annotations based on predictions by highest-performing algorithms (silver truth, ST), compared to human annotations (gold truth, GT); (iii) very valuable analyses of generalizability of algorithms, which are important for (existing and new) machine-learning (ML) based approaches; and (iv) a general shift (e.g., as indicated by numbers of new submissions) towards ML-based segmentation and tracking.

The combination of the above factors makes the manuscript an important and timely contribution to capture the state of the art as well as recent developments in biological image analysis. Due to the setup of the CTC, it relies on objective evaluations of methods using quantitative criteria, having the potential for an unbiased review that can address two (divergent) types of questions the readership may be interested in, namely (i) which segmentation and tracking methods to use or to further develop to tackle specific biological problems, and (ii) what the state of the field is more globally, and to what extent recent developments such as ML-based methods have contributed to improving it.

The manuscript aims to address both types of questions, but it appears more successful in addressing detailed questions of type (i) than more global questions of type (ii). A revision should, in particular, aim to leverage the wealth of data generated during the past years of the CTC by a more extended analysis. Important aspects of comparisons of human and machine performance and their relations to

GT annotations need clarifications. Finally, the manuscript is less self-contained (in the sense of being accessible to a non-specialist reader) than ref. 9. Detailed suggestions for addressing these aspects are given below.”

We would like to thank the reviewer for his/her positive comments about the quality and timeliness of our manuscript. We acknowledge that our analysis was mainly focused on the individual performance of methods and would benefit from a more global, quantitative analytic approach, taking advantage of the significant amount of available data. We have done such a global analysis to the best of our knowledge, as described in our response to the following comments:

“Major comments:

(i) Data analysis: The various iterations of the CTC as well as the expanding datasets have created a large dataset that is only explored in a qualitative manner. More formal analyses using simple statistical methods (e.g., ordinal or logistic regression) could exploit this data and help address the following questions more quantitatively; they seem important for the past and future development of the field (and are partially already discussed in the manuscript):

(i.a) Are dataset qualities and algorithmic performances (data in Table 1 and raw data for all algorithms) correlated, and if so, which features of datasets determine the difficulty of segmentation and tracking tasks? Answers could both help biologists gauge the success rate of analyzing a particular type of experiment, and developers focus future efforts.”

Indeed, studying the correlation between data quality and algorithmic performance would help biologists to predict what could be the expected outcome of the application of state-of-the-art algorithms to their data -based on their similarity with the properties of our existing datasets- and to optimize their imaging or experimental setup, minimizing the effect of the properties that correlate with poor outcomes. Furthermore, it should also help developers to determine where the efforts need to be focused to advance in the solution of the segmentation and/or tracking problem for each specific dataset.

We have analyzed the question of which dataset properties make the segmentation and tracking tasks more difficult in a new **Results** section of the manuscript, **Analysis of the relationship between image quality and algorithm performance**. This section describes the relationship between the main technical performance parameters (SEG and TRA) quantified for submissions above a reasonable quality threshold, and the dataset quality parameters listed in **Table 1**.

Since our repository is relatively small -we have only 40 unique independent values of each dataset characteristic (2 videos per dataset)- and highly heterogeneous within and between modalities, a multivariate analysis would not produce statistically significant results, and might result in misleading conclusions. Therefore, we focused on the effect of individual parameters. This was done by calculating the Spearman's rank correlation between algorithm performance and quality values. The analysis was done per sequence, at two levels of complexity: globally for all available datasets, and per data modality. Finally, specific results have been extracted based on per-quality parameter boxplots of the ranked dataset's performance values. The complete set of results can be found in a new **Supplementary PPT file 1a**. We refer the reviewer to the new **Results** section for the results of the analysis, that are also commented on in the **Discussion** section. There, we highlight the fact that there is only one parameter that can be globally considered influential for all types of modalities, as this relationship is mostly dictated by the modality, and in some cases, it is dataset-specific. Therefore, we have identified a set of modality-specific quality parameters that are correlated with the algorithm's performance. Furthermore, this analysis has helped us to realize that even if we consider that our dataset repository is large and comprehensive, it can be improved by including a higher number of datasets, possibly with different acquisition settings, to extend the conclusions extracted from this work. To facilitate the new results and discussions texts, we switched the coloring scheme used in **Table 1** from using three color classes to using gradual color scales. That way the quality level of each dataset is more accurately visualized, and the similarity -or dissimilarity- between datasets is better comprehended.

“(i.b) Are specific techniques (classification in Fig. 2b) related to algorithmic performance? Analysis results could support the conclusion on ML-based methods replacing traditional methods more quantitatively.”

As suggested by the reviewer, we have analyzed how different segmentation and tracking strategies, as well as individual detection, segmentation, and linking techniques, affect the technical performance of the benchmarked algorithms. This has been done by focusing on a pairwise comparison of traditional non-ML-based and ML-based techniques. We applied a non-parametric Mann-Whitney test when mutually comparing two distributions of technical performance scores, and a non-parametric Kruskal-Wallis test with Dunn's post hoc tests when mutually comparing more than two such distributions. The complete set of results can be found in a new **Supplementary PPT File 2a**, described in a new **Results** section of the manuscript, **Comparison, and analysis of the evolution of the segmentation and tracking paradigms**, and summarized in the **Discussion** section. This new analysis highlights and dissects the recent trend in replacing traditional non-ML-based approaches with better performing ones based on ML.

“(i.c) Do trade-offs exist between technical (Table 1, Figure 3) and biological (Figure 4) performance and what determines biological performance of algorithms? Biological performance substantially lags technical performance, and it will be important for future developments of methods and / or the CTC

itself to delineate (and discuss) avenues for improving biological performance, e.g. by expanding on p.10, ‘However, as the methods are not directly optimized ...’

Following the reviewer’s advice, we have analyzed the correlation between the technical tracking (TRA) measure and the biologically inspired measures (i.e., CT, TF, BC(i), CCA) all related to tracking performance. The analysis results can be found in **Supplementary PPT file 3**, summarized in a new **Results** section, **Analysis of the correlation between technical and biologically inspired measures**, and commented in the **Discussion** section. This analysis was done for each sequence at two levels: globally for all datasets and also per data modality. Globally, we observe a strong linear correlation between TRA and TF, and a moderate, non-linear correlation between TRA and CT. The correlations between TRA and BC(i), and TRA and CCA are weak. A similar trend can be observed for the correlations per microscopy modality. These findings explain why the biologically inspired measures that are uncorrelated with the TRA measure -all except TF- lag behind in terms of performance. It also supports the need for a new tracking-only benchmark specifically aimed at improving the development of algorithms that optimize those biologically inspired measures (CT, BC(i), and CCA). A commentary on this line has been included in the **Discussion** section.

(i.d) Do technical and biological performance evolve over time incrementally with existing methods, or by introducing new types of (ML) methods? The Figs. S1 and S2 give indications of performance improvements per dataset, but not in this more detailed sense.

Indeed, a more detailed analysis of the technical performance evolution over the years has clearly helped us identify important milestones that coincide with the introduction of novel concepts followed in the field of cell segmentation and tracking. In particular, these include detection-driven cell segmentation using ML-based prediction of detection markers in 2019, self-configured neural networks in 2020, and multi-branch predictions later on. These milestones have been integrated into a new **Results** section of the manuscript, **Comparison and analysis of the evolution of the segmentation and tracking paradigms**. The detailed temporal evolution of technical measures between the years 2013 and 2022 can be found in a new **Supplementary PPT File 2b**. Please note that we have not conducted a similar study for biological measures because the currently benchmarked methods have not been optimized toward their biological performance. Instead of presenting possibly biased conclusions, as already mentioned, we plan to establish a future tracking-only benchmark specifically aimed at improving the biological performance of cell segmentation and tracking algorithms, as commented in the **Discussion** section.

“(ii) Human vs machine performance: Definitions such as p.5 state that reference annotation quality is ‘measured as inter-annotator variability for GT’. It is unclear how this variability was determined (e.g., if based on binary decision problems such as existence of a link in tracking, for three annotator values

such as those reported for TRA_hum in Table 1 are not possible). Importantly, the presentations of human performance in Table 1 and Fig. 5) suggest comparability of human and algorithmic measures, which by the definitions (measures of variance vs measures of overlap with or transformability to human-derived GT) are not plausible. Please clarify definitions of human measures and clearly indicate / discuss comparability or non-comparability of measures in the text and captions.”

We have tried to clarify the definition of the quality measures of GT and ST in the heavily revised **Online Methods** section **Quality of annotations and human level performance**. We have also moved the first mention of these measures down in the manuscript, as they were probably difficult to understand before the quality measures used to compare the algorithms had been defined. A link to the **Online Methods** section is also included in the new **Results** section **Analysis of the relationship between reference annotation quality and algorithm performance**. To avoid confusion, we have also moved the annotation reference quality measurements (previously in **Table 1**) to a new **Table 2**. Furthermore, we have revised and clarified the text in the **Reference annotations** section and the mentioned **Online Methods** sections. Finally, we have renamed the quality measures to simplify understanding of what they are. We hope it is now easier to understand how these measures are computed and used.

Regarding the comparability of the measures, it is correct that the original text seemed to indicate that standard deviations and average values were compared, which would be incorrect. We hope that the new text will help clarify this. In summary, all quality/performance comparisons -between algorithms, between human annotations-, are based on the **average performance measures**. In turn, these measures are based on the overlap (SEG) or graph distances (TRA) between the participants' submissions and the consensus average of the manual annotation (comparison between algorithms) or between each individual human annotation and the consensus average of the manual annotations (human measures). The standard deviation of the reference annotations (GT) is only used as a reference of the variability of the manual annotations for a given dataset to determine the relevance of an algorithm's performance. For instance, they determine the minimum quality necessary for the new analyses presented with this revised version, or to set up a threshold for the existing generalizability study. In those cases, we do not compare the algorithm performance with the standard deviation of the annotation; we only determine if the algorithm performance is within the range of the **average quality annotation measure plus/minus a given number of standard deviations**.

Finally, as implied in this reviewer request, we have complemented the new analysis of the correlation between the quality of the datasets and the algorithm performance with a new analysis of the correlation between the quality of the reference annotations and the algorithm performance. This analysis is located in a new **Results** section named **Analysis of the relationship between reference annotation quality and algorithm performance**, which summarizes the correlative data included in the new **Supplementary PPT File 1b**. As could be somewhat expected and is summarized in the **Discussion** section, the performance of the algorithms follows the quality of the reference annotations, both for

the gold standard, and in an even stronger way, for the silver standard reference annotations. This indicates that what is complex to analyze for the human is also complex to analyze for the machine, highlighting the benefits of the increased coverage provided by the silver annotations and of the improved procedure of creating them. It also points to the need for incrementing the coverage of the annotation provided for some datasets and increasing the number of datasets for some modalities.

“(iii) Accessibility: In contrast to ref. 9, the present manuscript does not introduce much of the field’s terminology, or of the technical approaches, to a non-specialist reader. It may not be feasible or desirable to expand the manuscript accordingly, but I suggest careful revision to avoid unexplained technical terms as far as possible. For example, a sentence such as (p. 9/10) ‘The main contribution of this work consists of a new regularization term for the Cross-Entropy loss function based on Youden’s statistic, specifically designed to cope with highly imbalanced classes and weak annotations¹³.’ will be impenetrable for most readers of Nature Methods. A revision may consider more high-level descriptions, shortening the discussion of individual algorithms, or moving method details to SI text.”

We thank the reviewer for pointing this out. We have made an honest effort to simplify all technical descriptions -including those of the top-ranked algorithms- while maintaining the necessary level of rigor. We acknowledge that most readers of Nature Methods do not have a background in image processing. Still, some may have it, especially the potential readers of this manuscript, which targets not only potential users but also developers of cell segmentation and tracking algorithms. We hope that we have found the right balance and the manuscript is now more accessible to everyone. As suggested by the reviewer, the advanced reader also has frequent pointers to more technical, comprehensive descriptions in the form of supplementary information or references.

“Minor comments:

(i) Abstract: ‘These improvements include the creation of a new segmentation-only benchmark ...’. Given the lower performance of tracking compared to segmentation, do you consider a similar tracking-only benchmark for future CTC iterations?’

We thank the reviewer for this very relevant suggestion. Indeed, we are planning and already working on a new tracking-only benchmark that will incorporate the biological performance measures, such as the detection of complete tracks (CT), the correct detection of mitotic events (BC(i)), or the accuracy of the detected cell cycles (CCA). This will help the developers focus their optimization efforts on aspects of the problem of high relevance for some biological applications -cell lineage accuracy- that do not require a perfect definition -i.e., segmentation- of cell contours. Furthermore, as both Reviewers #1 and #2 emphasize the slow development of tracking techniques in comparison to segmentation techniques, we have added a paragraph in the **Discussion** trying to explain the reasons for this delay in

the biological measures and emphasizing the need for the above-mentioned tracking-only benchmark that could help improve the behavior of future algorithms.

“(ii) p.2, last para, ‘Notably, the state-of-the-art U-Net architecture was among the top-performing approaches for cell segmentation¹⁰.’: ref. 10 supports U-Net, not its evaluation in the CTC (ref. 9). In addition U-Net architectures were only in the top performers for selected (contrast-enhanced) data sets, according to ref. 9.”

The reviewer is correct. We have moved the ‘10’ right after U-Net. We have not replaced ‘10’ with ‘9’ as that whole paragraph refers to the conclusions of reference 9.

“(iii) p.3, ‘mesoscopic-size videos’ – please define the term.”

The term has been defined upon its first appearance.

“(iv) Table 1 and general: Avoiding abbreviations in annotations when possible (e.g., column labels) and adding structuring annotations (e.g., imaging mode or cell type for rows, dataset properties vs performances) could help the reader comprehend the complex data presentations more easily.”

We concur with the reviewer that the amount of data presented might be, at times, complex to comprehend due to the synthetic way it is presented in both tables and figures. At the same time, expanding the abbreviated and structured titles would force us to make the font smaller to a point that would be barely visible, or to enlarge those manuscript elements to a point where they might end up occupying too much space. Therefore, we have carefully revised all figure and table captions to ensure they are comprehensive enough by listing all abbreviations and structures used. Also, we have avoided the abbreviations “fus” and “hum” in the former Table 1 (now Table 2) and, instead, used measures properly defined in Online methods.

“(v) p.5, ST and GT: Please give indications of the quality of the ST annotations compared to GT annotations (for the subset of cells / tracks with overlap between the two), which is important to evaluate results of the generalizability study regarding training on GT, ST, and GT+ST.”

We believe that this should be clear now after our reply to Major point (ii). Indeed, the quality of the ST annotations, in regard to segmentation and detection, i.e., SEG_{GT} and DET_{GT} are calculated only in the subset of cells that overlap between ST and GT. However, please note that the GT and ST annotation quality measures cannot be directly compared and are not compared in the manuscript, as they are differently generated. The GT annotation quality measures indicate how far individual manual annotations are from their majority-based consensus, thus assessing the complexity of reaching the majority agreement, while the ST annotation quality measures are calculated directly against the GT.

“(vi) Fig. 2a: It would be informative to add the number of submissions for the different tasks, to give an indication of the problems the field is focusing on.”

We thank the reviewer for this suggestion. Figure 2A has been updated accordingly.

“(vii) p.10, generalizability study: Please explain OP_CSB at first use.”

OP_CSB is now defined at first use.

“(viii) p.10, ‘The “comparable score” criterion was defined as differing at most by the standard deviation of the human annotations for a particular dataset.’ Definition is unclear, please see major comment (ii) above.”

The text has been clarified. Based on this new version and our response to Major comment (ii), we hope that the text is clearer now.

“(ix) p.11, selection of unseen data types: Please discuss (ideally with the measures in Table 1) to what extent / in which aspects these ‘data types’ (rather: datasets) differed from the training data.”

Based on the reviewer’s suggestion to replace “data types” with “datasets,” we must emphasize that the generalizability study has two different aspects that need to be discussed separately. On the one hand, the use of the individual (GT, ST, GT+ST) or all (allGT, allST, allGT+allST) training configurations analyzes how augmenting the training set for a given dataset by using its “own” ST or the ST of all the datasets improves the performance of the methods compared to the use of only its “own” dataset GT. In this regard, it is assumed that the quality of the training data of a given dataset is almost identical to the test data of that same dataset, and the “all” training configurations contain a heterogenous pool where some datasets may have some level of similarity and some may not.

On the other hand, we analyze how the models, trained using the six configurations above, can generalize when segmenting unseen **data types**. We changed the nomenclature here from datasets to data types to emphasize that the training datasets of these unseen data types (or datasets) have never been seen by the models. The **Generalizability study** section now more clearly discusses the capacity of the models to generalize based on the similarities of the unseen data types and those of the (those) datasets used to train the models. As now explained in that section, the unseen datasets used are mostly different from the datasets used to train the methods. That is possibly the most likely cause of the poor generalizability found: Fluo-C2DL-Huh7 is only similar to Fluo-C2DL-MSK (in fact the scores obtained for the unseen data type is not far from the own obtained on the “known” data type), and the two

simulated datasets are also -being synthetic- of different nature from all other datasets. This has led to the conclusion that the methods tested are not able to generalize to data that significantly differ from the one used for training. Further research could be done in the future using unseen datasets that have similar characteristics to a higher number of the existing datasets, possibly particularizing the study by using only those similar datasets for training.

“(x) p.13, ‘an easy way to easily ... easily.’ – Please revise.”

The phrase has been revised.

“(xi) p. 16, last para of discussion: Please comment on biological relevance / open tasks as well (see major comment i.c).”

As indicated in our reply to Major comment (i.c) and Minor comment (i), the final part of the **Discussion** has been extended, explaining why tracking performance lags behind segmentation performance, especially regarding measures of biological relevance. We also confirm that we are actively working on a new tracking-only benchmark targeting new methodologies that may be optimized against the biologically inspired measures.

“(xii) Data availability: To strengthen reproducibility of the present analysis, providing a static dataset in a separate repository (instead of links to the CTC challenge web site, which will be changing) seems mandatory, specifically for raw data in figures and tables, and for method parameters.”

We are not convinced that this is necessary, as all the raw data for figures and tables are included in the Supplementary XLS files provided in this manuscript, and all the analyses are fully reproducible. The only links to the challenge website point to the entry pages for datasets and participants, which should stay the same in the future. This includes the references to the algorithm description -and parameters- which will always be identifiable by the group name. In the unlikely case that it evolves, it will be accompanied by a history of changes that will allow us to trace them back to the ones used in this manuscript. However, should the editorial board find it necessary, we will provide a static dataset in a separate repository.

“Reviewer #2:

Remarks to the Author:

GENERAL COMMENTS

Cell tracking challenging (CTC) is a competition aiming for boosting cell tracking techniques in 2D and 3D time-lapse images. This report has described the new data, performance criteria, algorithms, and

test results for CTC since the last report in 2017. In comparison to 2017, more datasets for training and testing, including those of large sizes, were added. A new criterion focused solely on cell detection was added. The accuracies of these new submitted algorithms in segmentation and tracking on various test datasets were compared. The top-ranked algorithms with high accuracies across many datasets were discussed in terms of performance and techniques. Some of the top-ranked algorithm's generalizability was evaluated on both images with seen datatype and images with the more challenging unseen datatype. The re-usability of some of the algorithms have also been discussed. This report revealed the advancements in cell tracking techniques and the recent trends in cell tracking methods related to the deep learning techniques. The descriptions of the top-ranked algorithms and their generalizability/re-usability will assist software developers and biologists users in selecting appropriate tracking methods to analyze the growing number of bio-medical images. I believe these new analyses are exciting and should be of great interest to a wide range of readers. On the other hand, I recommend that authors conduct additional analysis to assist readers in more easily selecting the appropriate techniques.”

We very much thank the reviewer for acknowledging the interest in the analysis presented in our manuscript. By addressing several major comments of Reviewer #1, which are very much in line with what this reviewer suggests, we believe the manuscript is now more helpful for potential readers and can facilitate their selection of appropriate techniques.

“MAYOR COMMENTS FOR THE ALGORITHMS ANALYSIS:

The current manuscript lacks an analysis of the relationships between image properties and proper tracking techniques. One of the most critical implications of this paper is that it can help developers and biologists choose appropriate methods for tracking cell of specific image datatype. Given that the authors have listed the dataset properties (Table 1) and classified the segmentation/tracking techniques used in each submitted algorithm (Fig. 2B), it would be much more interesting if the authors could further analyze the relationships between the dataset properties and the appropriate techniques.

For example, the results show that no method can achieve the best performance across all datasets (Fig. 3 and 5). This indicates that some techniques must be more appropriate for certain image properties, while others may be more appropriate for different image properties. Such relationships should be analyzed. Furthermore, if any image properties remain a significant obstacle to all methods, they should be described.

We thank the reviewer for these comments, which are in line with several comments of Reviewer #1. To address them, as explained at length in our responses to that reviewer, we have quantitatively, and statistically analyzed the relationship between the algorithm performance and the properties of the

datasets listed in **Table 1**. We have also analyzed the relationship between those performance values and the quality of the reference annotations listed in the new **Table 2**. This has been done both globally (see our response to Major comment (*i.a*)) and by segmentation, and tracking paradigms (see our response to Major comment (*i.b*)). This way, besides what we already had -i.e., the user could know the specific algorithm that can be optimal for a given new dataset, based on the similarity between their datasets and the ones provided by the challenge- we now explain also the reasons -based on the dataset properties- why specific datasets are more or less difficult to analyze. This should help a potential user to optimize the acquisition protocol, and highlight which techniques have more chances of success for a new data type, thus helping the user to choose between existing strategies, and the developers to optimize their new algorithms.

“Fig. 2B shows that some techniques, such as U-Net for detection/segmentation and Overlap/Distance based techniques for tracking (linking), are more popular than others. Do these popular techniques produce more accurate results on some or all datasets? Are there any techniques that are less widely used but produce better results in some or all datasets?”

As explained in our response to Reviewer #1, Major comment (*i.b*), we have indeed analyzed not only how frequently each technique is used (Figure 2B) but also which techniques are globally and individually -per dataset- more successful. The complete set of results of this analysis can be found in a new **Supplementary PPT File 2a**. Their discussion is available in a new **Results** section of the manuscript, **Comparison and analysis of the evolution of the segmentation and tracking paradigm**. In addition, a paragraph has been added to the **Discussion** section, explaining why, in our opinion, the U-Net is the most popular, and successful choice among other existing neural network architectures.

A more general question that covers questions above will be: in what scenarios/image properties each technique excel? Although the authors have mentioned the use of specific techniques when describing the top-ranked algorithms, a more systematic analysis will provide readers with more solid information for selecting appropriate segmentation/tracking techniques.”

As explained in our response to Reviewer #1, Major comment (*i.b*), and in addition to what is mentioned in the previous point, we have identified the dataset features that make tracking cells more complicated. This analysis correlates the techniques listed in Figure 2B with the dataset quality measures listed in Table 1. The complete set of new data produced can be found in **Supplementary PPT file 1a**. Its analysis is included in a new **Results** section of the manuscript, **Analysis of the relationship between image quality and algorithm performance**. Furthermore, as implied by Reviewer #1, and explained in our response to the particular comment (Major comment (*ii*)), we have included in this analysis the study of the correlation between algorithm performance and the quality of the reference annotations. This is explained in a new **Results** section, **Analysis of the relationship between reference annotation quality and algorithm performance**, which summarizes the correlative data

included in the new Supplementary PPT File 1b.

“MINOR COMMENTS FOR THE ALGORITHMS ANALYSIS:

In the discussion, authors emphasized the big change since 2017 is the use of deep learning techniques, especially in cell detection/segmentation. The data in Fig. 2B is consistent with this conclusion. The question is: are there any big advancement in tracking (linking) techniques, which should also be analyzed and described. If not, is it because the tracking algorithm is already mature and do not need large improvements? Or is it because the tracking (linking) part is relatively easier on the datasets used in this competition and do not need more advanced techniques?”

We thank the reviewer for this comment. As explained at length in the discussion and our response to Reviewer #1, Major comment (*ii*), the significant advance in the segmentation performance seen during the last years, mostly due to the increased use of deep learning models, has not been accompanied by a similar advance in the solution of the tracking part of the problem. Interestingly, there are already deep learning-empowered optical flow (i.e., pixel prediction) methods that are being applied in other areas of image analysis but have yet to make their way into cell tracking. This might be partly motivated by the fact that the tracking performance of the existing methods, calculated mostly based on their technical (TRA) performance, is comparatively higher than their segmentation counterpart. It is also motivated by the overwhelming presence of methods that address the segmentation and tracking aspects separately. This approach normally leads to an emphasis on the optimization of the segmentation and reduces to a residual post-processing step the linking part. We do hope to cause a change in the field by starting a new benchmark that will focus not only on the technical but also on the biologically inspired measures, which are clearly lower than the technical ones for all datasets. This will help developers to tune and optimize their next tracking development to these biological measures, which are of high interest in many real experimental settings. These ideas are covered in the **Discussion** section of the manuscript.

“MINOR COMMENTS FOR THE DATASETS:

1. New datasets have been added to CTC since 2017. I agree that increasing the number of datasets helps to better test the algorithms' performance in a variety of scenarios. However, the authors should explain the reason and the results of adding these data. For example, are these new datasets biologically important, and if so, why? Is it difficult to track some of the new datasets using previous methods? If so, why were they previously difficult to track, and why can new methods now track them?”

We thank the reviewer for this comment. The reasons for adding the new datasets differ for each new one. However, globally considered, it can be argued that adding datasets of different sources and modalities contributes to the diversity of the available training material. This is the case for most of the datasets added since 2017, which mostly represent modalities or applications not or only partially represented in the previous existing repository: Light Sheet Microscopy (LSM) for the analysis of the development of embryonic datasets, high-resolution real and synthetic confocal microscopy for the analysis of highly protruding migrating cells, brightfield microscopy for the analysis of the proliferation of stem cells. This diversity is especially relevant in the context of the dominance of machine learning models, the reason being two-fold. On the one hand, state-of-the-art deep learning algorithms offer alternative solutions to the generation of training data problems, as modeling of the cell appearances can be learned in the latent space in a data-driven fashion. Indeed, suppose a decent amount of certain types of cell images are already captured and properly annotated. In that case, it is possible to augment the dataset by training a latent diffusion model for image synthesis. The positions of cells over time can be described and specified by a known separate biological model. Then they can be fed into the latent diffusion model as the 'generative seeds' to render the appearances of the cells of the same phenotype as the annotated ones in an iterative fashion. On the other hand, when only a handful of cell images captured by "exotic" techniques are available, if adequate videos of the same or similar cell types have been captured by a different imaging modality, rendering an augmented "exotic" dataset could be achieved by training a style transfer generator (GAN). This GAN training can best preserve the cell appearance fidelity and temporal coherence of inter-frame motions when coupled with a proper spatial-temporal discriminator.

Furthermore, the new datasets are indeed biologically relevant. This was already mentioned in the **Discussion**, relative to the large mesoscopic embryonic datasets imaged by LSM. They are heavily used in developmental biology, and also at a smaller scale to study the normal and pathological activity of physiological systems -e.g., neurological, cardiac-, and to study mechanisms for wound repair, mechanobiology, etc. Regarding the real and synthetic videos of cancer cells with actin-stained protrusions, the applications are related to the study of the mechanobiology of mesenchymal cell migration, which is relevant in the context of the study of cancer, wound repair, etc. As for the mouse hematopoietic and muscle cells grown within hydrogel microwells, they constitute an efficient, high-throughput model to study the effect of proteins of the microenvironment, i.e., the niche, on the fate of stem cells. This is very important for another very active area of research as it is the study of stem cell biology. Finally, on a different note, each of these new datasets poses specific challenges for segmentation and tracking, not completely contemplated in the existing ones, as it is the size and -relatively- low resolution of the LSM embryonic samples, the specific interest for accurate cell-boundary delineation in the case of highly protruding cancer cells, or the high mitotic activity of the mouse stem cells grown in hydrogels. The **Discussion** section has now included a summarized version of this reply.

“2. Despite the name cell tracking challenge, the datasets used by CTC are essentially limited to cells in culture or embryos, according to the descriptions on the CTC website. Although these types of cell images have been studied for many years and have been the primary target for developing tracking methods, they do not represent all cell image types. New methods for tracking cells in mature organs such as brains and hearts in alive animals [1, 2], which have fewer/no divisions but much larger rotational/deformational motions than cells in culture/embryos, have emerged in recent years. The authors should make it clear to the readers whether the goal of CTC is to track cells during development or to run a competition on any type of time-lapse cell image. If the latter is the goal, it may be more appropriate to state unequivocally that the dataset used in the current competition is limited and should be expanded in the future.

REFERENCES:

- [1] Nguyen, J. P., Linder, A. N., Plummer, G. S., Shaevitz, J. W. & Leifer, A. M. Automatically tracking neurons in a moving and deforming brain. *PLoS Computational Biology* 13, (2017).
 [2] Wen, C. et al. 3DeeCellTracker, a deep learning-based pipeline for segmenting and tracking cells in 3D time lapse images. *eLife* 10, (2021).”

We thank the reviewer for this important comment. It is indeed true that the CTC dataset repository is exclusively made of videos of cells in 2D and 3D cultures or small animals in embryonic stages. As the reviewer indicates, this only covers some of the possible scenarios where cell tracking might be required. As he/she correctly points out, many applications involve tracking of cells *in vivo* in intact animals using non-invasive imaging techniques (e.g., intravital microscopy, optical or photoacoustic imaging) or *in vitro* within organoid models. These applications have specific tracking requirements due to the noisy and artificial nature of the images and the characteristics of the cell movement. Even if, now, these types of datasets have yet to be incorporated, they could be included in the future. A reduced version of this discussion has been included in the **Discussion** section.

“Reviewer #3:

Remarks to the Author:

Thanks to the development of microscopic techniques, there is a need to quantify image phenomena, which is logically preceded by the need to segment the image parts - cells. The paper is a logical follow-up to a similar report published five years ago, when a group of authors summarized the results of segmentation pipelines on realistic and often difficult datasets. The current update of this report reflects the progression in the field over the last 5 years, in particular the dominance of deep learning algorithms (and the pitfalls associated with them) and the proliferation of new microscopy techniques.

The current publication substantially follows the proven and rigid architecture and methodology of the 2017 Cell tracking challenge paper the history of the Challenge. The paper is mostly well understandable and because of the mentioned continuity, it is largely pointless to contradict the analytical procedures defined earlier. From a biologist's point of view, I am also very positive about the use of biological metrics for tracking. Now, in the current year, it is more about the urgent need to summarize the advances done in last 5 years. This review is therefore more concerned with the clarity and readability of the text.

The article reports evident progress in image segmentation. It is concluded that at least some aspects of tracking/segmentation are considered solved. Also, big improvement is observed on datasets considered difficult-to-track back in 2017. The CTC team increased the spectrum of datasets, so it better covers the spectrum of currently used methods, also including extremely challenging embryonal tomograms. The emphasis of this year's summary is mostly associated with deep-learning-specific issues, that is generalizability and problems with manual annotation. The paper is well written, but, in particular, the "new" sections would benefit corrections/clarifications."

We thank the reviewer for his thorough analysis of the manuscript, his mostly positive comments about its content, and his suggestions on how to improve the clarity and readability of the text. Based on his comments, an effort has been made to clarify points that were not properly explained, some of them described in our response to his questions below.

"Interestingly, for the generalizability, the authors report that the improvement obtained when increasing the complexity of the training data is small, but on the other hand, still, the performance on unseen dataset is dramatically lower. This is an interesting observation in the perspective of the fact that two of these three datasets are simulated (N2DH-SIM+ and N3DH-SIM+). Can this fact play a role? These simulated datasets are similar to some of the real ones in the CTC (therefore, why the scores are not higher?). At the same time, the simulated data potentially degrades the results by introducing artificially generated images. Could the authors comment that"?

We thank the reviewer for his comment. If we analyze the case of Fluo-C2DL-Huh7, indeed, the performance of the methods on this unseen dataset is substantially lower than the best-trained method. Interestingly, if we look at the behavior of the methods in the most similar dataset, based on the modality, staining and image quality (Fluo-C2DL-MSD), this dataset is also the one that seems less benefited from including other, different datasets in the training set. Furthermore, the performance of the models on unseen data (Fluo-C2DL-Huh7) is similar to that of the models trained with complex, heterogeneous data when analyzing Fluo-C2DL-MSD. This follows the fact that most of the features of both datasets are similar (see Table 1), and also more importantly, quite different from the other datasets. In the case of the synthetic datasets, as the reviewer suggests, the explanation could be related

to the synthetic nature of the datasets, which, even if visually resembling real datasets, are indeed different in terms of their texture and simplified motility. This is indeed confirmed by the comparison between the properties of these datasets and their most similar, modality related, real datasets. This is now explained in the **Generalizability study** section and commented in the **Discussion** section.

In the generalizability section the assessment on the unseen dataset is of great interest to readers, however, some sentences are misleading, could the authors please choose different words or clarify the following: “Regarding the performance of these methods,... we can appreciate a similar behavior, where the scores obtained stay far from the best-performing methods...”

We thank the reviewer for his recommendation. As explained in the previous section, we have heavily revised this section. The phrase referred by the reviewer is not in the manuscript anymore, and has been replaced by a longer, hopefully clearer one.

“In the end of the introduction, also the aspect of the network interpretability is mentioned, however, this is not discussed in the next sections of the article.”

We have indeed mentioned interpretability, as one of the challenges associated with using machine learning models. This is, however, something that relates more to the use of these models in tasks of predictive nature than to their use for segmentation and tracking tasks, which is the aim of the CTC. Therefore, we have not analyzed the interpretability of the submissions, as it is clearly out of the scope of the challenge. We have removed “interpretability” from that sentence to avoid an incorrect interpretation.

“Could the authors please comment the segmentation and tracking performance of simulated datasets (also in relation to the note on the SIM dataset usage in the generalizability section)? Specifically, the OP for both segmentation and tracking is slightly lower compared to somehow comparable N2DH and N3DH datasets. Is there any chance the lower OP is given because the simulated nature of the datasets, or is this effect purely random?”

When looking at the performance of the methods on the synthetic datasets (Fluo-N2DH-SIM+ and Fluo-N3DH-SIM+), it is correct, as the reviewer points out, that the top-performing methods produce worse results on these datasets than on the most similar datasets, Fluo-N2DH-GOWT1 and Fluo-N3DH-CHO, respectively. This difference in OP is due to differences in the segmentation (SEG) performance, as the tracking results are almost identical. Explaining these differences is complex, because of the high level of heterogeneity within and between datasets. However, we can try to provide some answers, based on the new correlation analysis done between the datasets properties and the performance of all methods (see the response to Reviewer #1 Major comment (*i.b*)). We can see that even if the main parameter (Ove), which globally correlates with algorithm performance, is very

similar for both the 2D and 3D datasets, some small differences appear in other modality-specific properties that could explain the differences we see in the performance. For the Fluo-2D datasets, the real dataset has, on average, slightly better resolution and more uniformly shaped cells than the simulated dataset. In the case of the Fluo-3D dataset, besides a higher resolution, the real dataset has better spacing (Spa) and more irregular shapes, both of which in Fluo-3D datasets correlate with better performance. It is possible that other cofactors may also play a role in the complexity, but these are the ones that are better aligned with our new analysis.

“One of the aspects of the CTC advances is the creation of a silver truth based on the intelligent combination of the best-submitted results; the procedure is understandably explained in the „Reference annotations“ section, however, in the first mentions, that is abstract, introduction and Table 1 legend its description is difficult to understand and the reader needs to jump between the online parts of the article. A short explanation in the introduction or the choice of other words might clarify how the silver truth was created and explain the term silver standard with better words.”

We agree that the caption of **Table 1** was not the correct place to define new concepts. First of all, we have removed it from the legend of **Table 2**, as the properties of the annotations, formerly part of **Table 1** are now listed in **Table 2**. The issue of clarity of the definition of the measures has been addressed in reply to Reviewer #1 “We have made an effort to clarify the definition of the quality measures of GT and ST, done in the now heavily revised **Online Methods** section **Quality of annotations and human level performance**. We have also moved down in the manuscript the first mention of these measures, as they were probably difficult to understand in the place where they were mentioned, before the quality measures used to compare the algorithms were first mentioned. A link to the mentioned **Online Methods** section is now included in the new **Results** section **Analysis of the relationship between reference annotation quality and algorithm performance**. Also, to avoid confusion, we have removed those reference quality measurements from **Table 1** and included them in a new **Table 2**, which is also presented in that same section. Furthermore, we have revised and clarified the text in the **Reference annotations** section and the mentioned **Online Methods** sections. Finally, we have renamed the quality measures to simplify understanding what they are. We hope that now it is easier to understand how these measures are computed and used.” Finally, to clarify the concept of creating silver truth, we have shortly expanded the phrase, which now reads: “the creation of a novel silver truth, i.e., a new set of reference annotations based on the intelligent combination of the best-submitted results, to dramatically increase the limited coverage of annotations included in the manual gold standard annotations”

“Last three columns of the table 1, following the previous comment on the definition of the silver standard, it is difficult to imagine what is being compared against what (gold vs silver standard). Please consider creating a schematic illustration or adding text explanation.”

We agree that including the quality parameters of the gold and silver standard annotations in Table 1 was probably not a good idea. It was done as those parameters can also relate to the performance of the methods, similar to the dataset quality parameters. We have extracted those columns and created a new **Table 2**, located at the end of the **Reference annotations** section, after the way these parameters were calculated. The description has also been expanded to make the text more understandable, as requested by Reviewer #1 as well.

“There is some inconsistency between Table 1 and the subsection “reference annotations” in the names used: inter-annotator variability, match with GT for ST and SEGhum, SEGfus, TRAhum in the table. Unification would be beneficial to understand”

We believe that this concern should also be addressed by the changes explained in our previous reply, with the given pointer to our response to Reviewer #1.

“The datasets used for segmentation-only task are the same as those used for tracking, that is, a timelapse of consequent frames which share a lot of visual similarities between each other. Then, if two subsequent frames are then used „one-for-training and one-for-validation“ (are they?), algorithms can achieve falsely better results. Can the authors how the frames for a segmentation task were handled, how (or if) this phenomenon exists and/or matters?”

We are not completely sure that we understand the reviewer’s question. We can clarify that the frames are not used “one-for-training and one-for-validation”. In all datasets, two videos are available to be used for training. They are different from the ones that are used for evaluation purposes.

It is true -perhaps this is what the reviewer wants to clarify- that some algorithm uses their tracking results to post-process their segmentation results, taking advantage of the fact that, indeed, two consecutive frames of the same video have a lot of spatial overlap. This is not a problem at all. Quite the opposite, it is a good practice by some groups, which tends to produce better segmentation results. However, when it comes to separating training from testing, it is always the case that the training datasets are different from the test datasets used to evaluate both the segmentation and tracking performance of the submissions.

“The section „Analysis of the technical performance of the submitted algorithms“ describes the architecture of the top-performing methods. As a biologist, this is outside the scope of my expertise.

The last decade has witnessed a significant expansion of native cell microscopy techniques in biology beyond fluorescence or intensity. Properties of light - polarization, phase, scattering, or spectroscopic techniques like Raman or Brillouin microscopy and many others are being used by biologists. While not all of them achieve the robustness of datasets necessary for learning networks, the need to

track/segment exists for them as well. Therefore, another challenge for the future is to work with very small datasets or with more "exotic" techniques. For generalizability, these might be even bigger challenges than learning simulated fluorescence datasets. How can this be reflected in future challenges? Please consider this more as a comment, I understand that the subject of the manuscript is different."

We thank the reviewer for this comment and acknowledge the existence of many exotic and not that exotic modalities that still need to be covered by this challenge (see our response to the last comment by Reviewer #2). We hope to be able to expand the CTC dataset repository, including some of these yet not covered datasets. However, as also explained in our response to the first Minor comment by that same reviewer, one of the advantages of the widespread use of deep learning is the existence of synthetic generators, also based on deep learning models. The synthetic generators can produce datasets of similar characteristics from a small annotated dataset. We copy here the relevant part of our response, which has also been included in the **Discussion** section of the manuscript: "On the other hand, when only a handful of cell images captured by "exotic" techniques are available, if adequate videos of the same or similar cell types have been captured by a different imaging modality, rendering of an augmented "exotic" dataset could be achieved by training a style transfer generator (GAN). When coupled with a proper spatial-temporal discriminator, this GAN training can best preserve the cell appearance fidelity and temporal coherence of inter-frame motions". In summary, if those same cells are available in larger quantities using a different modality, GANs can be used to generate synthetically augmented training sets. They incorporate the image texture properties of the exotic modalities into simulated cells that have the dynamics learned from cells stained using less exotic modalities.

"Minor

comments:

Table 1, description of the columns would benefit in mentioning the abbreviation prior each explanation, e.g. "SNR, Signal-to-noise-ratio; CR, contrast ratio,..."

Done, as suggested, in the caption of Table 1

"Fig. 3 the abbreviation "OP" for overall performance should be explained in text or figure legend. In the current form the abbreviation is explained just in supplementa."

Done, as suggested, upon its first use in the ***Analysis of the technical performance of the submitted algorithms*** section.

“Fig. 1d, Fluo-C3DH-A549, representativeness of the figure: are majority of cells isolated like shown in the representative fig., or is a proportion of cells touching?”

Yes, all the cells in this dataset are isolated. The emphasis of the analysis is placed on the accurate segmentation of the long cell protrusions more than on the actual tracking of the cells, which might be far from trivial in the case of touching cells.

In supplementary figures 1 and 2, please consider adding the dataset legend to the x-axis of the chart instead of mentioning it in the figure legend.

We thank the reviewer for this suggestion. The dataset names have been added to the x-axis of the chart.

“In the section “Analysis of the technical performance of the submitted algorithms”: please explain the meaning of “the CSB highlighted datasets”

The phrase tries to explain that in the CTB benchmark, the SEG scores can be improved in the subset of datasets, being the same that had been mentioned when referring to the SEG scores in the CSB benchmark at the beginning of the same paragraph. We have slightly modified the phrase to make it more clear.

“Supplementary methodology, Quality of annotations and human level performance, it is mentioned that the quality control criteria is present also in supplementary table with BIO measures. However, compared to CSB and CTB excel no such list in BIO excel is present.”

We thank the reviewer for pointing out this omission. Similarly to **Supplementary XLS Files 3 and 4**, the revised version of **Supplementary XLS File 5** now contains the “Quality Control Criteria” sheet as well.

“Section generalizability, something is wrong with the sentence „It also highlights the importance that those methods be provided with clear training instructions to make them usable for new types of data.”

We have replaced it with a clearer phrase: “It also highlights how important it is to provide clear instructions to train those methods in order to make them usable for new types of data”

Decision Letter, first revision:

Dear Carlos,

Thank you for submitting your revised manuscript "The Cell Tracking Challenge: 10 years of objective benchmarking" (N METH-AS50051A). It has now been seen by the original referees and their comments are below. The reviewers find that the paper has improved in revision, and therefore we'll be happy in principle to publish it in Nature Methods, pending minor revisions to comply with our editorial and formatting guidelines.

We are now performing detailed checks on your paper and will send you a checklist detailing our editorial and formatting requirements within the next two weeks. Please do not upload the final materials and make any revisions until you receive this additional information from us.

TRANSPARENT PEER REVIEW

Nature Methods offers a transparent peer review option for new original research manuscripts submitted from 17th February 2021. We encourage increased transparency in peer review by publishing the reviewer comments, author rebuttal letters and editorial decision letters if the authors agree. Such peer review material is made available as a supplementary peer review file. Please state in the cover letter 'I wish to participate in transparent peer review' if you want to opt in, or 'I do not wish to participate in transparent peer review' if you don't. Failure to state your preference will result in delays in accepting your manuscript for publication.

ORCID

Sincerely,
Rita

Rita Strack, Ph.D.
Senior Editor
Nature Methods

Reviewer #2 (Remarks to the Author):

The authors have addressed my previous concerns in a satisfactory manner. The revised manuscript provides abundant information, enabling users and developers to effectively evaluate and choose the appropriate cell segmentation and tracking algorithms.

Reviewer #3 (Remarks to the Author):

This version of the article brings a major revision of the text and tables of the article. The supplementary figures have also undergone significant changes. From a biologist's point of view, I highly appreciate the effort the authors put into the revision of the chapter "Analysis of the relationship between image quality and algorithm performance" (along with the extension of the accompanying graphs in the supplement).

Thanks to the changes made to the manuscript, the clarity, readability and comprehensibility of the text have been substantially improved.

The authors have responded precisely to all my previous comments. I therefore have no further comments on the text.

Final Decision Letter:

Dear Carlos,

I am pleased to inform you that your Analysis, "The Cell Tracking Challenge: 10 years of objective benchmarking", has now been accepted for publication in Nature Methods. Your paper is tentatively scheduled for publication in our June or July print issue, and will be published online prior to that. The

received and accepted dates will be August 5, 2022 and April 13, 2023. This note is intended to let you know what to expect from us over the next month or so, and to let you know where to address any further questions.

Once your paper is typeset, you will receive an email with a link to choose the appropriate publishing options for your paper and our Author Services team will be in touch regarding any additional information that may be required.

Please note that *Nature Methods* is a Transformative Journal (TJ). Authors may publish their research with us through the traditional subscription access route or make their paper immediately open access through payment of an article-processing charge (APC). Authors will not be required to make a final decision about access to their article until it has been accepted. [Find out more about Transformative Journals](https://www.springernature.com/gp/open-research/transformative-journals)

Your paper will now be copyedited to ensure that it conforms to Nature Methods style. Once proofs are generated, they will be sent to you electronically and you will be asked to send a corrected version within 24 hours. It is extremely important that you let us know now whether you will be difficult to

contact over the next month. If this is the case, we ask that you send us the contact information (email, phone and fax) of someone who will be able to check the proofs and deal with any last-minute problems.

If, when you receive your proof, you cannot meet the deadline, please inform us at rjsproduction@springernature.com immediately.

Once your manuscript is typeset and you have completed the appropriate grant of rights, you will receive a link to your electronic proof via email with a request to make any corrections within 48 hours. If, when you receive your proof, you cannot meet this deadline, please inform us at rjsproduction@springernature.com immediately.

Once your paper has been scheduled for online publication, the Nature press office will be in touch to confirm the details.

Once your paper has been scheduled for online publication, the Nature press office will be in touch to confirm the details.

Content is published online weekly on Mondays and Thursdays, and the embargo is set at 16:00 London time (GMT)/11:00 am US Eastern time (EST) on the day of publication. If you need to know the exact publication date or when the news embargo will be lifted, please contact our press office after you have submitted your proof corrections. Now is the time to inform your Public Relations or Press Office about your paper, as they might be interested in promoting its publication. This will allow them time to prepare an accurate and satisfactory press release. Include your manuscript tracking number NMETH-AS50051B and the name of the journal, which they will need when they contact our office.

About one week before your paper is published online, we shall be distributing a press release to news organizations worldwide, which may include details of your work. We are happy for your institution or funding agency to prepare its own press release, but it must mention the embargo date and Nature Methods. Our Press Office will contact you closer to the time of publication, but if you or your Press Office have any inquiries in the meantime, please contact press@nature.com.

To assist our authors in disseminating their research to the broader community, our SharedIt initiative provides you with a unique shareable link that will allow anyone (with or without a subscription) to read

the published article. Recipients of the link with a subscription will also be able to download and print the PDF.

Nature Portfolio journals [encourage authors to share their step-by-step experimental protocols](https://www.nature.com/nature-research/editorial-policies/reporting-standards#protocols) on a protocol sharing platform of their choice. Nature Portfolio 's Protocol Exchange is a free-to-use and open resource for protocols; protocols deposited in Protocol Exchange are citable and can be linked from the published article. More details can found at www.nature.com/protocolexchange/about.

Please note that you and any of your coauthors will be able to order reprints and single copies of the issue containing your article through Nature Portfolio 's reprint website, which is located at <http://www.nature.com/reprints/author-reprints.html>. If there are any questions about reprints please send an email to author-reprints@nature.com and someone will assist you.

Best regards,
Rita

Rita Strack, Ph.D.
Senior Editor
Nature Methods